# Diamond/GaN HEMTs: Where from and Where to?

**DOI:** 10.3390/ma15020415

**Published:** 2022-01-06

**Authors:** Joana C. Mendes, Michael Liehr, Changhui Li

**Affiliations:** 1Instituto de Telecomunicações e Departamento de Eletrónica, Telecomunicações e Informática, Universidade de Aveiro, 3810-193 Aveiro, Portugal; 2W&L Coating Systems GmbH, Bingenheimer Str. 32, D-61203 Reichelsheim, Germany; m.liehr@wl-coating.com (M.L.); c.li@wl-coating.com (C.L.)

**Keywords:** diamond, GaN, HEMT, thermal management, GaN-on-diamond, CVD

## Abstract

Gallium nitride is a wide bandgap semiconductor material with high electric field strength and electron mobility that translate in a tremendous potential for radio-frequency communications and renewable energy generation, amongst other areas. However, due to the particular architecture of GaN high electron mobility transistors, the relatively low thermal conductivity of the material induces the appearance of localized hotspots that degrade the devices performance and compromise their long term reliability. On the search of effective thermal management solutions, the integration of GaN and synthetic diamond with high thermal conductivity and electric breakdown strength shows a tremendous potential. A significant effort has been made in the past few years by both academic and industrial players in the search of a technological process that allows the integration of both materials and the fabrication of high performance and high reliability hybrid devices. Different approaches have been proposed, such as the development of diamond/GaN wafers for further device fabrication or the capping of passivated GaN devices with diamond films. This paper describes in detail the potential and technical challenges of each approach and presents and discusses their advantages and disadvantages.

## 1. Introduction

Gallium nitride (GaN) is a wide bandgap compound III-V semiconductor with high breakdown electric field, high electron mobility, and high electron saturation velocity that translate in a tremendous potential for high power and high frequency applications (Table 1). The GaN high electron mobility transistor (HEMT) is a device that takes advantage of the two-dimensional electron gas (2DEG) that spontaneously forms at an aluminum gallium nitride (AlGaN)/GaN heterojunction thanks to the strong internal piezoelectric and spontaneous polarization. This 2DEG typically exhibits high values of sheet carrier density (≈10^13^ cm^−2^) and carrier mobility (1000–2000 cm^2^/(V⋅s)) [1].

GaN HEMTs can be used in power switching for information technologies, automotive, healthcare, and industrial manufacturing applications [1,2,3]. Thanks to the large bandgap, leakage currents in GaN power devices are orders of magnitude smaller than in silicon (Si), allowing for operation at higher temperature without thermal runaway and reducing the cooling requirements. The high breakdown electric field allows shorter drift distances for a given blocking voltage, when compared to Si devices, yielding to a drastic reduction in the specific on-resistance that in turn translates into smaller device area and correspondingly lower capacitance. This reduces switching losses and enables higher switching frequencies.

GaN HEMTS have also paved their way into mobile and satellite communications and radar systems [4]. In addition to the properties listed above, the high breakdown electric field of GaN allows higher matching impedances and circuits with broader bandwidth and high power-added efficiency (*PAE*) [5]. The ability of GaN to withstand higher temperatures further increases the power density of a given HEMT device and power amplifiers with absolute power levels of tens to hundreds of Watts have been reported [6].

Despite the fact that commercial HEMT devices based on a different combination of processes and design technologies are currently available from a broad range of manufacturers [4], GaN-based technologies still face some challenges that affect their overall performance and limit their potential benefits.

Due to the intrinsic nature of the 2DEG, GaN HEMTs are normally-on (depletion mode) devices. For power switching applications, normally-off devices are preferred due to static power consumption, simplification of circuit design, and safety concerns. Normally-off transistors can be obtained with different techniques [14], however their performance is typically worse than that of their normally-on counterparts [15,16].The existence of electrically active surface traps located at the passivation/top-layer interface [17] and of bulk traps present in the GaN and buffer layers [18] induces effects such as current collapse [19,20,21], dynamic on-resistance (or knee walkout) [22], degradation of cut-off frequency [23], and DC-RF dispersion [24], which compromise the reliability of the devices and prevent harnessing the full potential of GaN HEMT power devices [25].Due to the intrinsic nature of GaN HEMTs, harsh and localized self-heating in the conducting channel may occur [26]; this effect increases with the device power density and further compromises reliability [22,27]. On one side, the electrical behavior of the traps mentioned in the above paragraph is temperature-dependent [25]. On the other side, additional phonon scattering in the channel degrades the 2DEG effective carrier mobility, leading to degraded DC and RF performance [28]. Finally, since the relation between the mean time-to-failure (MTTF) of an electronic component and its operating temperature is semi-exponential [29], even a small temperature reduction can have a great impact on the lifetime of HEMTs with thermally-activated degradation mechanisms [30].

From what was described above, it can be concluded that the capability of efficiently transferring the heat away from the localized hotspots and the consequent control of the device temperature is fundamental to achieve high levels of stability and reliability in HEMT applications [31]. Diamond has the highest thermal conductivity (κ) of any bulk material, and the integration of diamond films and GaN HEMTs as substrates or packaging has already proven to enhance the extraction of the heat generated during the devices operation, leading to a substantial decrease in the junction temperature as well as to an increase in the maximum power density the HEMTs can safely handle. This anticipates a superior high-frequency handling capacity, higher energy efficiency and flexibility, and a better utilization of the electromagnetic spectrum.

Diamond has been successfully integrated with HEMT devices following different approaches. This manuscript aims at describing each of them in detail, pointing out the technical challenges and benefits, and providing the reader with a critical discussion of the feasibility of each approach. The manuscript is organized as follows: Section 2 discusses the critical aspects that impact the thermal management of GaN HEMTs; Section 3 describes the different strategies that have been followed by different groups to integrate diamond and GaN into high performance devices; Section 4 describes the challenges faced by each integration technology and discusses their feasibility; Section 5 draws the main conclusions and Appendix A summarizes the performance of the different GaN/diamond HEMTs reported so far.

## 2. Thermal Management of GaN HEMTs

### 2.1. GaN HEMT

The concept of mobility enhancement through modulation doping of an aluminum gallium arsenide (AlGaAs)/gallium arsenide (GaAs) multilayer heterojunction was introduced by Dingle et al. in 1978 [32]. Since the energy of the GaAs conduction band is lower than the energy of the AlGaAs donor states, electrons from the later move into the GaAs regions, forming a 2DEG. By introducing a rectifying contact on top of the heterojunction, Mimura et al. [33] were able to control, by field effect, the concentration of the 2DEG. Soon after the deposition of high quality GaN films on sapphire substrates by Metalorganic Chemical Vapor Deposition (MOCVD) [34] and using the same principle, the AlGaN/GaN HEMT was reported by Khan and co-workers [35]. In a conventional AlGaN/GaN HEMT, the current flowing in the 2DEG channel between the source and drain Ohmic electrodes is modulated by a negative bias applied to the gate Schottky contact.

The cross-section view of a general GaN HEMT structure is shown in Figure 1 and the κ of the materials typically used are listed in Table 2. The actual structure, composition, and thickness of each layer in a particular HEMT depend on its specific purpose and/or the vendor’s manufacturing practices. Since the review and discussion of the literature presenting different devices modifications is out of the scope of this work, only the layers considered relevant to the thermal transport are represented in Figure 1. Other possible layers (not shown) would include the spacer and cap layers.

Substrate. Homoepitaxial growth of GaN-based films is hampered by the limited availability of GaN substrates in standard wafer sizes. As a consequence, the different layers are typically deposited by either Molecular Beam Epitaxy (MBE) or MOCVD onto sapphire, Si, or silicon carbide (SiC) substrates. Epitaxial films with dislocation densities of 10^8^ cm^−2^ are typically obtained [36]; dislocation densities lower than 10^7^ cm^−2^ involve hydride vapor phase epitaxy (HVPE) growth.Nucleation layer. The deposition of high quality epitaxial GaN films with smooth surfaces and low dislocation density is not a straightforward task due to lattice mismatch and to the difference in the coefficients of thermal expansion (*CTE*s) of GaN and substrate [37]. A nucleation layer, typically 40–200 nm of aluminum nitride (AlN) [38], is thus initially deposited on the substrate surface for strain accommodation and increased interface resistivity [39,40].Strain relief layer. AlGaN/GaN transition layers, often up to 1 µm thickness, further accommodate the lattice mismatch during the growth of GaN on the foreign substrate [41].GaN layer. A 0.6–1.5 µm-thick GaN buffer layer that provides electrical isolation to reduce substrate leakage and prevents the propagation of threading dislocations and contaminants that might migrate from the substrate into the top high quality channel region follows.Barrier layer. The typically 5–25 nm-thick barrier layer can be made of AlN [42], indium gallium nitride (InGaN) [43], or other high bandgap alloys, but Al_x_Ga_1−x_N with an aluminum (Al) fractional content in the range of 22–32% [6] is the most widely reported material.Passivation layer. A thin dielectric layer, typically silicon nitride (Si_3_N_4_), compensates the surface/interface states responsible for the current collapse issue by introducing shallow donors [44].Field plate. The source and gate-connected field plates are usually employed to reduce the strength of the electric field near the gate terminal, reducing the gate tunneling injection current responsible for charging the surface traps [45].

### 2.2. Getting the Heat Out

Thanks to the high breakdown voltage and saturation velocity of carriers in the 2DEG, AlGaN/GaN HEMTs are able to handle substantial power densities, which result in self-heating and highly-localized power densities (in some cases as high as 10^5^ W/cm^2^) [49]. The resulting high temperatures decrease the MTTF, the performance, and the reliability of HEMT devices—and this makes the implementation of efficient thermal management techniques mandatory. This is particularly important for devices that have inherently small features yet process large power densities, such as high-power RF/millimeter-wave transistors and single-mode visible semiconductor lasers.

Most of the heat generated during device operation will diffuse from the hotspot through the different layers represented in Figure 1 until it reaches the heat sink attached to the back of the substrate. A few intrinsic obstacles hinder the transfer of the heat towards the heat sink. The AlGaN/GaN strain relief transition layers improve the electrical performance at the top of the GaN layer, however, if the concentration of Al is higher than 5%, the κ of the AlGaN decreases to around 1/10th that of bulk GaN, hampering the transfer of heat from the GaN buffer layer to the underlying substrate [41].

The thermal transport across the strain-relief layer/nucleation layer/substrate interfaces also plays an important role in determining the overall κ of the GaN HEMT material system. The existence of an interface between two solids results in the scattering of thermal energy carriers (electrons and phonons), which translates in the appearance of a thermal boundary resistance (*TBR*) between the different materials and in a temperature discontinuity across the interface [50,51].

Despite its low thickness in comparison to the GaN layer, the nucleation layer profoundly impacts the transport of heat from the strain relief layer towards the underlying substrate. Instead of being composed of high quality AlN, this layer contains dislocations, grain boundaries, and point defects (impurity atoms and vacancies) [52,53], within the itself or near the interfaces, that hinder heat transport by increasing phonon scattering rates and reducing the phonon mean free path [54].

The interfacial and nucleation layer thermal resistances contribute with an effective *TBR* between the strain relief layers and the substrate between 10 and 70 m^2^⋅K/GW [38] that may cause an additional 30–50% channel temperature rise in AlGaN/GaN HEMTs [38,55].

### 2.3. Why GaN-Diamond HEMTs?

In the search of good thermal conductors, carbon-based materials, such as highly oriented pyrolytic graphite (HOPG), graphene, and diamond, are obvious candidates.

HOPG is an anisotropic material with high in-plane κ (κ_in-plane_ ≈ 2000 W/(m·K)) but a much lower out-of-plane κ (κ_out-of-plane_ ≈ 6–9 W/(m·K)) [56]. Despite being also an electrical conductor, graphite already found its place for thermal management of electronic components at system level, and high κ graphite films for assembling integrated circuits or CPUs, for example, are available from a few vendors. Other examples include graphite heat sinks [57] and composite graphite/metal laminates [58].

Graphene is another anisotropic material with even higher in-plane κ (κ_in-plane_ > 3000 W/(m·K)) [59]. However, the particular value of κ depends significantly on the preparation method and can be reduced greatly up to one order of magnitude compared to that of pristine graphene because of poor alignment and structural defects in the material [60]. Due to its intrinsic 2D nature, electrically conductive graphene films are more suited for integration at device level. In 2012 Yan et al. [61] reported the use of graphene quilts for the thermal management of GaN HEMTs, obtaining a ≈20 °C decrease in the hot spot temperature in transistors operating at ≈13 W/mm.

Diamond is an isotropic material with high κ (2200 W/(m·K), increasing to 3300 W/(m·K) in the case of isotopically pure material) while being electrically insulating, with a breakdown field 60 times greater that of Si (2 × 10^7^ V/cm [13])—Table 1. Single-crystal diamond (SCD), grown by high pressure high temperature (HPHT) method, has the best thermal and electric properties; however its area is limited to a few mm^2^. Alternatively, polycrystalline diamond (PCD) films can be grown by chemical vapor deposition (CVD) on large-area substrates such as Si, overcoming the area limitation while still guaranteeing κ values in the range 1000–1800 W/(m⋅K) [62].

## 3. Integration of Diamond and GaN

The integration of diamond and GaN for the fabrication of HEMTs with superior thermal handling capability has been an active research topic involving academic and industrial institutions for more than 20 years. Generally speaking, the integration of both materials can be done in two ways: replacing the GaN substrate with diamond or placing the diamond on top of the device, close to the gate hotspot.

The fabrication of GaN-on-diamond wafers can be made using three fundamentally different approaches: (i) depositing diamond films by CVD directly on the back of GaN wafers, following the substrate removal (hereafter referred to as GaN-on-diamond); (ii) bonding GaN HEMT wafers and diamond substrates (bonded wafers); and (iii) growing the epitaxial GaN layers directly on diamond substrates (GaN epitaxy). Placing the diamond on top of the HEMT device can be done simply by growing the diamond films directly on the passivated surface of the device (capping diamond).

The historical evolution of each approach is presented in detail in the following paragraphs.

### 3.1. GaN-on-Diamond

The GaN-on-diamond concept was initially introduced in 2003 [49]. The original idea relied on the deposition of the PCD film directly on the back of the GaN layer. The first GaN-on-diamond wafer pieces were produced in 2004 by etching the Si substrate of a GaN wafer initially deposited by MOCVD, followed by depositing a 50 nm-thick dielectric layer, and then by growing a 25 µm-thick PCD film by hot filament CVD (HFCVD). The PCD was deposited on the Ga-face, leaving behind an N-face GaN-on-diamond wafer after the etching of the temporary Si carrier. The fabrication process remains fundamentally the same till today and is represented in Figure 2. The required steps involved (i) performing the GaN epitaxy on a Si substrate, (ii) bonding the GaN HEMT epilayers onto a temporary Si carrier, (iii) etching away the original host Si substrate, (iv) depositing a 50 nm-thick layer of SiN onto the exposed rear face of the GaN, and (v) depositing a 25 µm-thick PCD film by HFCVD onto the dielectric. Finally, by (vi) removing the temporary Si carrier, a free-standing GaN-on-diamond wafer was obtained.

By 2006, the process had been optimized to fabricate a Ga-face (i.e., right side up) GaN-on-Diamond HEMT epitaxial wafer and the operation of an AlGaN/GaN HEMT with a 25 µm-thick PCD film located 1.2 µm below the electron channel was reported [63]. The scanning electron microscope (SEM) cross-section of the GaN-on-diamond wafer is shown in Figure 3a. These first unpassivated HEMTs had a high contact resistance that translated in low current capability (maximum drain current *I*_D max_ = 306 mA/mm) and low peak transconductance (*g*_m peak_ = 70 mS/mm). Further iterations improved the performance of the HEMTs [64,65], nevertheless, and in spite of having half the thermal resistance *R*_th_ of GaN-on-SiC HEMTs (≈6 against ≈12 K⋅mm/W, respectively), in 2007 GaN-on-diamond HEMTs were still outperformed by GaN-on-SiC technology—Figure 3b. By 2009 GaN HEMTs with 75 µm of PCD showed cut-off (*f*_T_) and maximum oscillation (*f*_max_) frequencies of 85 and 95 GHz, respectively [66], and soon after that the first RF power amplifier module [67] was reported.

In 2011 DARPA introduced the Near Junction Thermal Transport (NJTT) concept that aimed at extracting the heat from within 1 µm of a transistor’s active region. The low κ AlN/AlGaN nucleation/transition layers under the GaN channel were eliminated [41,68,69], allowing the deposition of the films within a few hundreds of nanometers away from the hot junction, and the thermal conductance between the GaN and the diamond was more than doubled. Continuous wave (CW) load-pull tests performed at 10 GHz showed that the GaN-on-diamond HEMTs reached over 7 W/mm output power density (*P*_D_) and 46% *PAE* at 40 V drain bias (*V*_D_) [70]; however, and despite showing a 25% lower temperature rise for the same thermal power than their GaN-on-SiC counterparts, these devices had a high gate leakage current associated with residual surface defects in the gate region which still prevented the harnessing of the full potential of the technology.

The κ of the diamond films was further improved by replacing the HFCVD process with microwave plasma CVD (MPCVD) and by the end of the NJTT program, in 2014, the GaN-on-diamond technology had allowed the reduction of the junction temperature by 40–45% and the tripling of the areal RF power density in comparison with GaN-on-SiC [71].

At the same time the bottleneck of the heat extraction was recognized to be the *TBR* between GaN and diamond (*TBR*_GaN/diamond_) [72] and most of following research focused on decreasing it, whether by decreasing the dielectric thickness, by using a different dielectric, or by optimizing the diamond nucleation layer [10,72,73,74,75,76,77,78,79,80,81,82,83,84]. The impact of the thickness of the GaN buffer layer on the *R*_th_ of the HEMT devices [85,86,87,88,89,90] and the effects of the stress caused by the difference in the *CTE*s of GaN and diamond [91,92,93,94,95,96] were also evaluated by different research groups. A more detailed description and discussion of the main findings is included in Section 4.1. The mechanical and thermo-mechanical integrity of the diamond/GaN interface, which impacts profoundly the reliability of the devices, was also addressed [97,98,99]. As a general finding, Liu et al. concluded that the GaN/diamond interface has a high mechanical stability, showing the potential of this material system for the fabrication of reliable devices [97].

The performance of more recent HEMT devices was thoroughly analyzed by Ranjan and co-workers [28,100], who evaluated the effect of the bias conditions on the self-heating and transport properties of GaN-on-Si and GaN-on-diamond (with 30 nm of SiN) HEMTs. A ≈4 times improvement in the DC and RF performances of the later was observed. The DC *P*_D_ of GaN-on-diamond HEMTs was 27.56 W/mm for 55 V applied *V*_D_, whereas GaN-on-Si devices were burnt at ≈9 W/mm for 20 V *V*_D_. The reduction of *I*_D max_ due to channel self-heating for GaN-on-diamond and GaN-on-Si HEMTs was 10% and 33%, and for *V*_D_ = 10 V *f*_T_/*f*_max_ were 10.2/31.4 GHz and 7/18.2 GHz, respectively. The GaN-on-diamond HEMT had an almost constant small signal gain for *V*_D_ between 10 and 40 V. Finally, the increase rate of the gate current with *V*_D_ was 3.3 times smaller for GaN-on-diamond devices. Figure 4a–c show some of the devices’ electrical characteristics.

In 2019, HEMTs fabricated on latest generation Element Six GaN-on-diamond wafers (with a 30 nm-thick dielectric layer) showed 2.95 K⋅mm/W *R*_th_, 56 W/mm DC power capability, and average/maximum channel temperature of 176/205 °C [101]. Nevertheless, these devices had high leakage currents which ultimately limited their breakdown voltage, showing there is still some room for optimizing the fabrication process and maximizing the performance of GaN-on-diamond HEMTs. In the same year, researchers from RFHIC [102] reported the fabrication of 4″ GaN-on-diamond wafers with a *TBR*_GaN/diamond_ of 31.0 m^2^⋅K/GW and an uniformity of ±10%. The development of an inner slot via hole process allowed the opening of 10 µm-diameter holes in the diamond using a laser drilling process—Figure 5. On-wafer pulsed load-pull tests performed at 2 GHz revealed 18.1 W/mm *P*_D_ for an encapsulated 10 × 200 µm gate HEMT.

The evolution of the relevant technological parameters (dielectric material and thickness, diamond film κ, and value of *TBR*_GaN/diamond_) since 2006 is listed in Table 3. It can be seen that most of the GaN-on-diamond HEMTs fabricated by this method feature a ≈30 nm-thick SiN layer, and a *TBR*_GaN/diamond_ in the range of 20–30 m^2^⋅K/GW. Section 4.1 describes in more detail the impact of the dielectric layer on the experimental *TBR*_GaN/diamond_ values and the electrical parameters of GaN-on-diamond HEMTs fabricated so far using this method are summarized in Table A1 in the Appendix A.

### 3.2. Bonded Wafers

In 2013 BAE Systems proposed a “device-first” technology that allowed the placement of the diamond heat spreader within 1 µm of the device hotspot [107,108]. After the complete fabrication of the devices, the wafer was bonded to a temporary carrier and the substrate and the GaN buffer layer were removed. The back side of the HEMTs was then bonded at room temperature (RT) to a 1″ square PCD diamond substrate, fabricated in a different step, using an adhesive and pressing the two materials together. Thanks to the low roughness of the GaN back surface after the removal of the buffer layers (<1 nm), and depending on the κ of the adhesive used, *TBR*_GaN/diamond_ was estimated to be in the range 15−60 m^2^⋅K/GW. The generic process flow is schematically represented in Figure 6.

In 2014, functional GaN HEMTs originally fabricated in a SiC substrate were bonded to a 1″ PCD wafer at a temperature lower than 150 °C [109] by means of a 35 nm-thick layer of Si-containing bonding material [110]. The experimental value of *TBR*_GaN/diamond_ was 34 m^2^⋅K/GW and the yield of the bonding process was ≈70% (Figure 7a). Even when dissipating 3 times more power, the temperature of the GaN-on-diamond HEMTs was lower than that of their GaN-on-SiC counterparts. However, original GaN-on-SiC devices outperformed GaN-on-diamond ones: *I*_D max_ and *g*_m peak_ were reduced by 16% and 11%, respectively, after the GaN-on-SiC HEMTs were transferred to diamond. The degradation of the DC characteristics was attributed to changes in residual mechanical stress in the device epitaxial layers during the substrate transfer process, as well as to the mechanical and chemical treatments applied. The RF characteristics of the GaN-on-diamond devices also degraded in comparison with GaN-on-SiC ones: at 10 GHz and for *V*_D_ = 20 V *PAE*/*P*_D_ were 38%/3.4 W/mm and 48%/4.6 W/mm for both devices, respectively, when tuned for maximum power, and 42%/3.0 W/mm and 57%/4.1 W/mm (when tuned for efficiency). According to the authors, this was primarily due to the omission of air-bridge structures in the GaN-on-diamond devices with the unconnected device channels acting as RF parasitics during power measurements.

Upon solving these issues, the *PAE* of 12 × 50 µm GaN/diamond HEMTs increased to 51% and *P*_D_ to 11.0 W/mm at 10 GHz. For the same *V*_D_, the *P*_D_ of 4 × 50 µm GaN-on-SiC HEMTs was only 9.2 W/mm, showing a 3.5 times areal power increase with GaN-on-diamond HEMTs (Figure 7b) [111]. Even under these conditions, the temperature at the center gates was slightly lower for the GaN-on-diamond HEMT than for the GaN-on-SiC HEMT (195 against 202 °C, respectively). The main challenge of this process was identified as the ability to achieve large area bonding with very low *TBR*_GaN/diamond_ [110].

In 2017, Liu and co-workers [112] bonded HEMT devices previously fabricated on SiC substrates to a 3″ commercial PCD wafer at a temperature of 180 °C and obtained a functional device yield over 80%. The experimental *TBR*_GaN/diamond_ (51 m^2^⋅K/GW) was still relatively high and the DC current showed a 12−19% reduction due to self-heating; nevertheless, the peak junction temperature of a 10 × 125 µm HEMT with compressed gate pitch of 20 μm decreased from 241 to 191 °C after being transferred from the SiC to the diamond substrate, suggesting a 20% reduction in *R*_th_. CW load-pull measurements were performed at 10 GHz and class AB operation on 4 × 125 µm/40 μm gate pitch HEMTs; after being transferred to the PCD substrate, the same HEMT delivered 5.5 W/mm *P*_D_ with a *PAE* of 50.5% (against 4.8 W/mm and 50.9% when on the original SiC substrate).

The RT bonding of GaN and PCD films [113] or SCD substrates [114,115] has also been achieved using surface-activated-bonding (SAB). SAB is a direct solid state covalent bonding method that takes place in ultra-high vacuum conditions without obvious interfacial chemical reactions. The previous bombardment of the to-be-bonded surfaces with an argon (Ar) ion beam induces the surface activation, generating surface dangling bonds and making it possible to bond the surfaces at RT [116]. A few nm-thick Si interlayer is typically sputtered on the diamond and GaN surfaces to improve the adhesion. The two activated surfaces are then pressed together at RT and the dangling bonds form covalent bonds at the interface. Transmission electron microscope (TEM) images of the resulting uniform GaN/diamond interface can be seen in Figure 8a (from [113]). After the bonding process the Si layer was ≈24 nm-thick. The bombardment of the diamond surface with Ar ions induced its amorphization, creating an additional ≈3 nm-thick amorphous diamond layer. The thickness of the Si layer was further reduced to ≈10 nm [115] (Figure 8b), creating an interface with a *TBR*_GaN/diamond_ of ≈19 m^2^∙K/GW. In a slightly different experimental setup, the surfaces activation was performed with a mixed beam of Si and Ar ions; the thickness of both Si interlayer and amorphous diamond layer was reduced to ≈2 nm (Figure 8c) and the *TBR*_GaN/diamond_ was as low as ≈11 m^2^∙K/GW. Despite these interesting preliminary results, functional HEMTs on this GaN/diamond material stack are yet to be reported.

Using a similar SAB method with an intermediate Si nanolayer, in 2019 Mitsubishi Electric Corp. announced the successful transfer of a GaN-on-Si multi-cell HEMT to a SCD substrate [114]. No voids were identified in the ≈6 nm-thick bonding interface and the improved DC characteristics of the GaN-on-diamond devices showed that the GaN HEMTs layers were successfully transferred to the diamond substrate (Figure 9). This achievement is expected to improve the *PAE* of high-power amplifiers in mobile communication base stations and satellite communications systems, thereby helping to reduce power consumption. Mitsubishi Electric targeted the commercial launch of GaN-on-diamond HEMTS for 2025.

Wang et al. [117] reported bonding GaN and SCD/PCD diamond substrates at RT and in atmospheric air. Following the surface activation with Ar ions, a double molybdenum (Mo)/gold (Au) layer (5 nm/11 nm) was sputtered on the surfaces of both materials and they were pressed together with an applied load of 2000 N. The bonded surfaces showed a voidage as low as 1.5% and the bonding strength was evaluated as 6.8 MPa. Even though no thermal measurements were made, after 1000 cycles of thermal cycling between −45 and 125 °C the bonding area remained at 73%, suggesting that the Mo/Au nanolayer can effectively balance the difference in the *CTE*s of GaN and PCD wafers.

Minoura and his colleagues from Fujitsu Limited [118] bonded AlGaN/GaN and indium aluminum gallium nitride (InAlGaN)/GaN-on-SiC HEMTs to SCD substrates by a modified SAB method. A thin (<10 nm) titanium (Ti) layer was previously deposited on the surface of the SCD substrate to prevent the amorphization of the diamond surface during the bombardment with the Ar ions. The TEM image of the SiC/diamond interface is shown in Figure 10a. Using this method, the *TRB* at the diamond/SiC interface was 66 m^2^⋅K/GW and the *R*_th_ of the AlGaN/GaN HEMT bonded to the SCD substrate was about 1/3 compared to that without diamond. The *P*_D_ of InAlGaN/GaN HEMTs measured with pulsed load-pull measurements (for *V*_D_ = 50 V, a pulse width of 10 µs, and 10% duty cycle) increased from 14.8 to 19.8 W/mm with the bonding of the SCD heat spreader structure. With 1% duty cycle, the SCD-bonded HEMT showed a *P*_D_ of 22.3 W/mm. Figure 10b,c show the variation of the normalized *P*_D_ with the duty-cycle and the *P*_D_ as a function of *V*_D_, respectively.

SAB of GaN HEMTs and diamond was also achieved by sputtering a 450 nm-thick AlN layer on both GaN AlN nucleation layer (after removal of the Si substrate) and diamond surfaces, followed by surface activation with an Ar^+^-based plasma and by thermocompression at 160 °C [119]. The strain relief layers and the AlN nucleation layer prevented the flow of heat from the top GaN layer into the diamond substrate; as a consequence, the temperature of the HEMT on the diamond substrate was higher than that on the Si substrate [120].

Van der Waals (VdW) bonding, a process first employed for GaAs thin films [121], has also been used to bond GaN devices and SCD/PCD substrates at temperatures below 300 °C [122,123]. This technique guarantees a good thermal interface (*TBR*_GaN/SCD_ was estimated to be as low as 10 m^2^⋅K/GW [122]) without the observation of stress or degradation. RF-devices operating at 3 GHz with improved efficiency (*PAE* of 54.2% against 50.6% for GaN-on-Si HEMTs)) were demonstrated with SCD; on the other hand, the bonding obtained with PCD was not reproducible [123].

The evolution of the relevant technological parameters of GaN/diamond bonded wafers is summarized in Table 4. The *TBR* of the GaN/diamond interface obtained after the SAB and VdW bonding of GaN and SCD can be as low as 11 and 10 m^2^⋅K/GW, respectively. Functional HEMTs have been fabricated on GaN bonded to SCD and PCD substrates. The electrical parameters of HEMTs fabricated using this method are summarized in Table A2 in the Appendix A.

### 3.3. GaN Epitaxy

The last technique to fabricate GaN-on-diamond wafers involves the epitaxial deposition of GaN on the diamond substrate. However, the deposition of epitaxial GaN films on SCD on PCD substrates is inherently difficult, however, due to a series of factors [124]. First, the large lattice mismatch between the two materials (around 13%) can generate defects and lead to poor crystal quality. It should be mentioned, however, that the lattice mismatch between GaN and sapphire is even higher (16% [36]), so this fact alone is not determinant. In addition to the lattice mismatch, the large difference in the *CTE*s of diamond and GaN (0.8 × 10^−6^ K^−1^ against 5.6 × 10^−6^ K^−1^, respectively, whereas the *CTE* of sapphire is 7.5 × 10^−6^ K^−1^) can lead to a highly stressed interface that will further impact the dislocating density and lead to the possible cracking of the GaN layer. Finally, in the case of PCD substrates, the absence of a fixed epitaxial relationship between the GaN and the diamond increases the difficulty of nucleating a continuous epitaxial GaN layer.

Nevertheless, different groups have tried to grow GaN on diamond substrates. First reports date from 2003, when Hageman and co-workers [125] reported the growth of 2.5 µm-thick polycrystalline and hexagonal GaN films on (100) natural SCD by a dual step. A thin GaN layer was initially grown by MOCVD on a 10 nm-thick AlN nucleation layer previously deposited on the SCD substrate. Subsequently, the MOCVD pre-grown samples were used as templates for the growth of thick GaN layers using HVPE. The group later reported the growth of 0.07−1.55 µm-thick GaN films on (001) SCD HPHT substrates by MOCVD [126].

Using a different method, Dussaigne et al. [127] reported the growth of 1 µm-thick GaN epilayers on 100 nm-thick AlN layers previously deposited on (111) HPHT SCD substrates by ammonia (NH_3_)-source MBE. The films displayed a low RMS roughness of 1.3 nm and some cracks due to the difference in the *CTE*s of diamond and GaN; cracks were not formed for GaN epilayers with thicknesses lower than 250 nm. In a subsequent work [128], 200 nm-thick AlN and GaN strain engineered interlayers were deposited on top of the AlN buffer layer. An 800 nm-thick GaN layer with 8 × 10^9^ cm^−2^ dislocation density was then grown, followed by 24 nm of AlGaN (with an Al content of 28%). A 2DEG formed at the interface, allowing the fabrication of HEMT structures with *I*_D max_ = 730 mA/mm, *g*_m peak_ = 137.5 mS/mm, and *f*_T_/*f*_max_ = 21/42.5 GHz [129].

Following a different approach, researchers from NTT Corporation reported the formation of a 2DEG in an AlGaN/GaN heterostructure formed on a 600 nm-thick GaN layer epitaxially grown on a type Ib (111) SCD substrate by metalorganic vapor-phase epitaxy (MOVPE) [130] with similar dislocation density (8.4 × 10^9^ cm^−2^ [131]). The AlN/GaN stress-relief layers were deposited on a 180 nm-thick single crystal AlN buffer layer deposited on the diamond substrate and the fabricated AlGaN/GaN HEMTs showed *I*_D max_ = 220 mA/mm and *f*_T_/*f*_max_ = 3/7 GHz. The *R*_th_ of GaN-on-diamond HEMTs was 4.1 K⋅mm/W, against 7.9 K⋅mm/W for similar GaN-on-SiC structures. By depositing the GaN layer on type IIa (111) SCD diamond substrates, *R*_th_ was further decreased to 1.5 K⋅mm/W and RF operation with 2.13 W/mm output power density and 46% *PAE* was achieved [131]. The impact of the misorientation angle of the (111) SCD surface on the surface morphology of the HEMT structures was also studied by the authors [132,133].

Other groups reported growth of GaN on PCD and NCD substrates. The benefits of this approach would be twofold: large-area PCD substrates are available from a few vendors, and the price/area is significantly lower than the one of HPHT SCD substrates. However, only polycrystalline GaN films could be initially deposited. Van Dreumel et al. [134] deposited polycrystalline GaN layers by MOCVD on the surface of nanocrystalline diamond (NCD) films previously deposited by HFCVD on Si substrates, while Zhang et al. [135] deposited fine grained polycrystalline GaN films by MOCVD on the nucleation surfaces of freestanding PCD films prepared by glow discharge. The final goal in this last work was the fabrication of diamond/GaN surface acoustic wave devices, though, and not the fabrication of AlGaN/GaN HEMTs.

More recently, Webster et al. [124] reported the ability to grow epitaxially-oriented GaN films on thick PCD substrates by MOVPE. An AlN layer was previously deposited at 650 °C as nucleation layer for the subsequent growth of a 1.5 μm-thick GaN layer at higher temperature, on top of which the AlGaN/GaN HEMTs structures were fabricated. SiN stripes were then deposited on the AlGaN/GaN stack and the unmasked regions were etched down to the PCD substrate. After an epitaxial layer overgrowth (ELO) re-growth cycle the dislocation density of the GaN layer was reduced by two orders of magnitude (from ≈7 × 10^−9^ to < 10^8^ cm^−2^). With this technique the team was able to grow GaN with a significant degree of epitaxial orientation on an area up to 15 μm^2^ on a PCD substrate—Figure 11. The appearance of some cracks on the GaN surface was attributed to the large difference between *CTE*s of diamond and GaN.

Later the team proposed a different approach [136]; the (111) Si substrate from commercial diamond-on-Si substrates was etched, exposing the back surface of the PCD substrate that was shown to feature a thin Si_x_C layer formed during the deposition of the PCD on the Si substrate. According to the authors, this Si_x_C layer provides sufficient crystallographic information for the epitaxial growth process to occur. Following the etching of the Si substrate, the PCD films exhibited some curvature, due to the thermal stress caused by the high diamond deposition temperatures and the difference in the *CTE*s of both materials. AlN or Al_0.75_Ga_0.25_N nucleation layers were grown on the SiC layer and GaN was deposited by MOVPE. Interestingly, the growth on the nucleation surface of the curved diamond substrate was shown to effectively remove the surface defects induced by the non-single crystalline nature of the Si_x_C layer and to reduce the tensile stress induced by the *CTE* mismatch, allowing the growth of crack-free GaN epitaxial layers up to 1.1 µm thick. As the thickness of the GaN layers increased, the dislocation density was significantly reduced (Figure 12). Upon optimization of the whole process, the tensile strain inherent in GaN epitaxial layers grown on PCD can be further reduced, and thicker crack-free GaN layers can be deposited. Since the increase of the thickness of the GaN epitaxial layer is usually accompanied by reductions in the dislocation density, the use of thicker GaN layers is expected to have a positive impact on the performance of HEMT devices fabricated using PCD substrates.

In 2020 Ahmed et al. [137] also proposed the use of ELO to integrate GaN and PCD diamond; the processing steps are represented in Figure 13. Unlike the procedure described in [124], where the GaN was deposited directly on the PCD by MOVPE, the AlGaN/GaN stack capped with a thin SiN layer was initially deposited on a Si substrate by MOCVD. 500 nm-thick stripes of PCD were then selectively deposited on the SiN by HFCVD following the procedure described in [138]; the exposed SiN was then removed by reactive ion etching (RIE) and the GaN surface was made accessible (step 4 in Figure 13a). The wafer was then returned to the MOCVD reactor for the ELO of GaN (step 5 in Figure 13a). Figure 13b,c show the SEM images of the final structure before and after the regrowth step, respectively. A continuous GaN film formed across 5 μm-wide diamond stripes separated by 5 μm-wide GaN windows oriented along the $\langle 1\bar{1}00\rangle$ direction of the initial GaN. The dislocation density on ELO GaN (≈10^7^ cm^−2^) was ≈2 orders of magnitude lower than on initial GaN (≈10^9^ cm^−2^). Although coalescence was achieved, several voids, pinholes, and cracks were visible in some locations along the coalescence regions; the cracks were generated due to the release of thermal stress energy upon cooling of the wafer following GaN growth. In order to obtain the final GaN/PCD final devices, some steps are further required: MOCVD growth of the HEMT structure, removal of Si substrate and exposure of diamond strip.es surface, thick diamond CVD, and transistor fabrication.

The experimental results obtained so far are compiled in Table 5. As a summary one can say that research efforts focused on both SCD and PCD substrates, however functional HEMTs have only been fabricated on SCD substrates. Despite promising results, the interest on these substrates faded away and more recent research focused on the ELO of GaN films on PCD substrates with reduced dislocation density. The electrical parameters of HEMTs fabricated on GaN films deposited on SCD substrates are summarized in Table A3 in the Appendix A.

### 3.4. Capping Diamond

The capping diamond concept was initially proposed in 1991 [139]. This approach relies on the deposition of the PCD film directly on the device die, immediately on top of the passivation layers, bringing the diamond film in close contact with the hot spots—Figure 14. Initial calculations predicted a 50 °C reduction in the channel temperature of a 2 W/mm GaAs device caused by “thermally shorting” the source, gate and drain contacts with a highly thermally conductive diamond film [140].

Despite no experimental works reported the direct growth of PCD films on GaAs FETs, the concept was picked up by the GaN community and the first successfully diamond-coated working GaN device was reported 10 years later, in 2001, by researchers from the Fraunhofer Institute and from DaimlerChrysler high frequency electronics labs [46]. Diamond films with thicknesses between 0.7 and 2 µm were deposited directly on the SiN passivation layer of GaN FETs using an MPCVD ellipsoid reactor operating at 2.45 GHz at a temperature lower than 500 °C—Figure 15a. Since the regular ultrasonic seeding with diamond particles was found to damage the SiN protective layer, an alternate seeding method based on the sedimentation of fine diamond particles from an agitated emulsion was used. The output and transfer characteristics of the devices were measured before and after the deposition of the 0.7 µm-thick PCD film, showing that the coated FETs remained fully operational—Figure 15b.

Despite these encouraging first results, the next successful reports describing the deposition of diamond films directly on HEMT devices date from 2010 [141]. In a joint work by the University of Maryland and the Naval Research Laboratory (USA), the authors reported the deposition of a 0.5 μm-thick NCD film on a GaN-on-Si HEMT by MPCVD. A 50 nm-thick silicon dioxide (SiO_2_) passivation layer was previously deposited on the surface of the device and the deposition temperature was increased to 750 °C. The NCD-coated device exhibited 20% lower temperature in comparison with the SiO_2_-passivated one, however, after NCD coating, *I*_D max_ and *g*_m peak_ decreased from 176 mA/mm and 145.4 mS/mm to 157 mA/mm and 113.9 mS/mm, respectively. This effect occurred due to a reduction in pinch-off voltage following the increased drain-gate coupling through the SiO_2_ and NCD film. The NCD film was unintentionally contaminated with boron impurities (1 × 10^10^ cm^−2^) and an increase of the leakage current was observed. In order to protect the thermally sensitive Schottky gate contact, a “gate after diamond” process was further developed by the team [142]. This approach involved the deposition of a 0.5 μm-thick NCD layer with κ in excess of 400 W/(m∙K) on a 50 nm-thick SiO_2_ layer after completion of the mesa and ohmic processes, but before the gate metallization step. An oxygen (O_2_)-based plasma etch was then used to recess etch the diamond in the gate region before metal deposition. Similarly to what happened in the previous approach, *I*_D max_ decreased from ≈360 to ≈270 mA/mm after the deposition of the NCD film. The process was further optimized to allow the deposition of the NCD layer directly on the GaN surface [143,144], which improved the performance of the 2DEG in comparison with a reference HEMT passivated with SiN (2DEG density/mobility of 1.02 × 10^13^ cm^−2^/1280 cm^2^/(V⋅s) against 8.92 × 10^12^ cm^−2^/1220 cm^2^/(V⋅s), respectively), increased *I*_D max_ and *g*_m peak_ from 380 to 445 mA/mm and from 114 to 127 mS/mm, respectively, and decreased the device on-resistance from 14.6 to 11.9 Ω⋅mm. At 5 W/mm DC power, the temperature of the NCD-capped HEMT was ≈20% lower than that of the SiN-capped one and the *R*_th_ was 0.98 K⋅mm/W, ≈3.75 times lower. The use of a p-type diamond gate electrode was also proposed [145]. Again *I*_D max_ increased from ≈290 mA/mm (with a nickel (Ni)/Au gate) to ≈430 mA/mm (with the NCD gate), the on-resistance decreased from 29.4 to 12.1 Ω⋅mm and the leakage gate current decreased by nearly one order of magnitude at a gate voltage of −10 V. In a later process development, the SiO_2_ layer was replaced with a 10 nm-thick SiN layer [146]. The capping of a SiN-passivated GaN-on-SiC HEMT with 500 nm NCD lead to an ≈8% reduction in the peak temperature, less than the 20% obtained with the gate-after-diamond approach due to the SiN passivation layer and the resulting increased *TBR*_GaN/diamond_ [147].

Following a different approach in 2011 Alomari et al. [148] reported the deposition of a 0.5 μm-thick NCD film on In_0.17_AlN_0.83_/GaN HEMTs by bias-enhanced nucleation (BEN) and HFCVD at temperatures of 750–800 °C—Figure 16a. The HEMT structures included the dielectric passivation and stress control layers based on SiO_2_ and Si_3_N_4_ and a thin sputtered Si layer for BEN step (Figure 16b). The DC characteristics of the devices remained fundamentally similar after the NCD deposition. *f*_T_ and *f*_max_ were 4.2 and 5 GHz, respectively, and the RF-tested device showed high gate leakage. In 2014 the NCD layer thickness had been increased to 2.8 μm [149].

Zhou et al. [150] deposited 155–1000 nm-thick PCD films onto Si_3_N_4_-passivated AlGaN/GaN-on-Si HEMT structures and used transient thermoreflectance experimental results, together with device thermal simulations, to evaluate the impact of the diamond capping layer thickness on the maximum device temperature. A 12% maximum reduction in peak channel temperature could be achieved with a 16 × 125 µm/50 µm gate-pitch AlGaN/GaN-on-Si HEMT with a 1 µm-thick PCD film deposited on the source-drain opening. If the *TBR*_GaN/diamond_ was not included, a further 10% temperature reduction could be expected. Little further thermal benefit was predicted when using PCD films thicker than 2 µm, with only a maximum 15% temperature reduction. If PCD could be grown on both source-drain opening and metal contacts, a 1.5% better thermal benefit would be achieved for thicker films by increasing the area of the heat spreader. The quality of the initial few µm of the capping diamond layer was also shown to play an important role in reducing the channel temperature [151].

In 2019, researchers from the Power and Wide-Band-Gap Electronics Research Laboratory and Lake Diamond SA deposited 3 μm-thick PCD heat spreaders on the top of vertical GaN PiN diodes using a 30 nm-thick SiN adhesion layer [152]. With 0.9 W of dissipated power, the temperature of the coated device was 64% lower than the temperature of a reference uncoated device.

The appearance of biaxial strain in the Al(Ga)N layers due to the high diamond deposition temperatures, the lattice mismatch between the layers, and the different *CTE*s of the stack materials was studied in detail by Siddique et al. [91]. In their study, a 46 nm-thick SiN layer was used to protect the III-nitride layers beneath the AlGaN barrier layer during the diamond CVD. Figure 17 shows scanning transmission electron microscope (STEM) images of the SiN layer (a) before and (b) after the diamond deposition; even though the SiN layer was partially degraded during the deposition of the diamond and was thinned down to 20 nm in some regions, it remained continuous across the entirety of the AlGaN barrier layer. The deposition of the diamond film was seen to increase the biaxial tensile stress in the AlN and GaN layers by 3%, which caused a 4.5% reduction of the total 2DEG sheet charge density (from 1.04 × 10^13^ to 0.99 × 10^13^ cm^–2^). A 52.8 m^2^⋅K/GW *TBR*_GaN/diamond_ was measured (including the SiN, GaN cap, AlGaN, and AlN layers).

In order to prevent the modulation of the 2DEG due to the stress induced in the channel by the capping diamond layer, Arivazhagan et al. [153] proposed depositing the diamond layer on the drain electrode alone (instead of directly on the device channel). The impact of this approach was evaluated using thermo-electrical simulations. The results suggest that the deposition of the PCD film on the drain electrode will lower self-heating and allow higher *I*_D max_ than in conventional GaN-on-Si HEMTs—without inducing any change in the 2DEG sheet charge density or in the device threshold voltage.

Despite the technological difficulties inherent to this technological approach, a breakthrough has been recently achieved by Fujitsu Limited. In 2021 Yaita et al. reported a diamond-coated GaN-on-SiC HEMT with improved DC characteristics [154]. A 2.5 µm-thick PCD film was deposited by HFCVD on a 100 nm-thick SiN capping layer at 700 °C. In order to prevent the degradation by the elevated temperatures required to deposit the diamond film, the Schottky gate contact was replaced by a 40 nm-thick SiN insulating layer, followed by a Ni/Au gate contact. In addition, metal heat spreaders were attached to the deposited PCD. The *I*_D_ and *g*_m_ of the diamond-coated HEMT increased from 0.9 to 1.1 A/mm and from 102 to 148 mS/mm, respectively (Figure 18a). At the same time, for 25 W/mm *P*_D_ the hotspot temperature lowered by more than 100 °C, which corresponds to a decrease in *R*_th_ from 12.7 to 7.4 K⋅mm/W, and to a ≈40% reduction in the amount of heat generated by the diamond-coated GaN HEMT (Figure 18b). Fujitsu aims to commercialize improved-heat-dissipation GaN HEMT amplifiers in year 2022 for use in weather radar systems and next-generation wireless communication systems [155].

Experimental results presented so far have been limited to the scope of single or dual-gate HEMTs. Zhang and co-workers implemented 3D thermal simulations to evaluate the impact of capping multi-finger HEMTs with PCD layers [156]. They observed that the capping diamond layer reduced the junction temperature and the temperature non-uniformity in the near-junction region across the channel; the efficiency of the capping diamond layer was also observed to increase with increasing thickness but with a decreasing trend. The largest thermal benefit could be expected under challenging conditions, such as high *P*_D_, narrow gate pitch, high *TBR*, as well as for traditional GaN-on-Si HEMTs. In the particular case of a 12-finger GaN-on-diamond HEMT operating at of 6 W/mm *P*_D_ per gate, 20 µm gate pitch, and assuming a similar *TBR* between both GaN/diamond interfaces of 15 m^2^⋅K/GW, the inclusion of a 20 μm-thick PCD capping layer would reduce the junction temperature from 195.8 to 172.2 °C, which corresponds to a net decrease of 23.6 °C. By reducing the thickness of the PCD layer to 1 μm, the reduction of junction temperature would still be as high as 14.9 °C.

The impact of capping double-channel HEMTs was also evaluated with thermo-electrical simulations [157]. The authors observed that the PCD layer provides a lateral heat conduction path close to the hot spot located at near the drain side of the gate edge, modulating the channel lattice temperature distribution and making it become more uniform. For a *P*_D_ of 46 W/mm the peak lattice temperature was reduced by 64 °C, indicating that the PCD layer plays an important role in device heat dissipation. Similar to what was observed in [156], the effect of lattice temperature reduction increases with increasing PCD layer thickness. Taking the effect of temperature reduction and cost into consideration, the authors proposed the optimum thickness for the PCD layer to be 1 μm.

The impact of capping a pulsed-mode AlGaN/GaN HEMT with a PCD layer has been also evaluated with thermal 3D simulations [158]. The PCD capped layer not only reduced significantly the peak junction temperature but also suppressed its oscillation in the pulse mode operation, smoothing the temporal variation of the junction temperature. Again the cooling performance of the PCD capped layer was observed to increase with rising thickness but with a decreasing trend. The overall efficiency of this approach was shown to be more effective under harsh thermal conditions, including smaller duty cycle, higher TRB, and lower κ substrate.

The evolution of the relevant experimental results has been compiled in Table 6. While experimental results and functional HEMTs were reported between 2001 and 2014, most recent work focused on evaluating the biaxial strain induced by the diamond deposition temperatures (typically in excess of 500 °C) and on anticipating the thermal benefit of the diamond layer using thermal simulations. The electrical parameters of diamond-capped HEMTS are summarized in Table A4 in the Appendix A.

## 4. GaN/Diamond HEMTs: Where to Go?

Each of the previously described approaches has some advantages over the others and, simultaneously, shows room for improvement if particular technological developments can be achieved. This is will be the topic of the current section.

### 4.1. Challenges of Fabricating GaN-on-Diamond Wafers

The deposition of diamond films on the back of GaN wafers has been routinely performed by Element Six for more than 10 years. The fabrication of GaN-on-diamond wafers involves the deposition of 100 μm-thick diamond films at temperatures higher than 700 °C. The large deposition temperatures induce a large residual stress at the GaN/diamond interface that may cause the bowing of the wafer and the cracking of the GaN layers. In addition, the *TBR*_GaN/diamond_ and the low quality of the initial layers of the deposited diamond films are currently the bottleneck of the heat extraction. Each of these issues will be discussed in detail in the following paragraphs. 

#### 4.1.1. Decreasing Thermal Stress

One of the biggest issues of this approach is related with the high diamond deposition temperature (>700 °C) which induces a large residual stress at the GaN/diamond interface because of the mismatch between the *CTE*s of both materials [159,160]. The residual stress depends on the GaN thickness, on the diamond growth temperature, and on the sacrificial carrier wafer [91,92]. Residual stresses larger than 1 GPa at the free surface of the GaN have been reported [93]; these elevated stress conditions induce layer cracking and wafer bow, and impact the electrical performance of the devices [161,162,163,164]. A large thermal stress at the GaN-diamond interface causes a significant reliability concern when considering the function and lifetime of GaN-on-diamond devices.

To avoid the issues related with the residual stress, a slightly modified approach has been recently proposed [94]. The GaN was initially grown on an AlN nucleation layer deposited on a Si substrate. The Si substrate was selectively etched, leaving behind 0.5 mm-diameter GaN membranes with the AlN nucleation layer exposed onto which 50 µm-thick diamond films were further grown. The high quality diamond/AlN interface showed no visible voids or cracks, confirming the strong bond between both materials. This approach presents no technological barrier to the incorporation of an AlN “initiation” layer into the GaN buffer close to the device channel; however, due to the low thickness of the GaN membranes and the simultaneously high temperatures required for diamond growth, the bowing of the membranes during the diamond deposition step cannot be neglected. This phenomenon was recently analyzed in detail in [95] using commercial GaN-on-Si wafers as the starting material. The processing steps are described in Figure 19a. The GaN membrane diameter was shown to impact the maximum displacement from the original plane (bowing) before and after diamond deposition, as well as the membrane stress. The 5 mm-diameter membranes allowed a larger displacement with a subsequent lower stress value, however the larger bowing “pushed” the GaN surface further into the plasma during diamond deposition, exposing it to high temperatures and resulting in thermal runaway and damage to the GaN/III-N film—Figure 19b. In addition, the large membrane bows present a big challenge for device manufacturing using contact lithography. A possible way to prevent the bowing would be the use of pre-stressed GaN-on-Si wafers as the starting material.

Following a radically different approach, represented in Figure 20, Jia et al. [96] reported a low stress GaN-on-diamond wafer fabricated by dual sided deposition of diamond. In this process, the temporary carrier represented in Figure 2 was a 2 µm-thick Si layer followed by 100 µm of low-quality PCD deposited by CVD. A thin SiN layer was deposited on the surface after the substrate removal, followed by high-quality PCD. The final free-standing GaN-on-diamond wafer was obtained by removing the low-quality PCD and Si layers. The surface and quality of the GaN layer after the etching of the low quality PCD and Si layers remained the same while the stress was reduced to 0.5 GPa. However, it should be mentioned that the AlN buffer layer was not removed and this may have helped to reduce the internal stress caused by diamond growth.

#### 4.1.2. Optimizing the Thermal Barrier Resistance at the Diamond/GaN Interface

The minimization of the *TBR*_GaN/diamond_ will have a positive impact on the thermal performance and reliability of GaN-on-diamond HEMTs. In theory this can be achieved by reducing the thickness of dielectric layer, thereby reducing its bulk *R*_th_, or by improving the interface between GaN/dielectric and dielectric/diamond to reduce the interface effects. As an example, for a particular GaN-on-diamond HEMT, the reduction in the *TBR*_GaN/diamond_ related with the SiN layer from 13 to 3 m^2^⋅K/GW would reduce the total *R*_th_ of the device from 2.8 to 1.9 K⋅mm/W, which corresponds to a 35% decrease in the peak operating temperature rise at a given power level [73]. However, one should also keep in mind that, despite smoother GaN/SiN/diamond interfaces lead to lower *TBR*_GaN/diamond_, they also show reduced interfacial fracture toughness, in comparison with rougher interfaces [99], which could negatively impact the devices reliability.

The standard fabrication of GaN-on-diamond wafers requires a 30 nm-thick SiN layer to protect the surface of the GaN from the diamond deposition conditions. Despite its low thickness, this layer may contribute with a *TBR* of ≈30 m^2^∙K/GW, adding more than 20% to the total device *R*_th_ [74]. Some groups decreased the thickness of the SiN layer to 5 nm and obtained *TBR* values of 9.5 and 6.5 m^2^⋅K/GW [10,82] (close to the theoretical minimum of 5.5 m^2^⋅K/GW calculated by the diffuse mismatch model [82]).

An apparently simple way of decreasing *TBR*_GaN/diamond_ would be to remove the SiN layer. However, depositing diamond directly onto GaN is not a straightforward task. At typical diamond CVD temperatures (700 °C and above), the atomic hydrogen (H) can etch the surface of the GaN substrate (forming NH_3_ and liquid gallium (Ga)). This etching can be prevented if the density of diamond seeds on the GaN surface is so high that the lateral diamond growth rate exceeds the GaN etching rate. In this case, a protective diamond layer grows to cover the GaN surface before significant etching can occur. However, even under these conditions the interface between GaN and diamond is rather weak because Ga does not form a carbide. This means the diamond adheres to the GaN surface mainly via weak VdW interactions, rather than by strong covalent bonds. This becomes a serious problem when the coated samples are cooled down to RT due to the difference in the *CTE*s of both materials, which causes compressive stress to accumulate in the diamond layer and can lead to delamination of the entire diamond layer [75].

Despite these difficulties, different groups have reported direct growth of diamond films on GaN [10,82], however the experimental values of *TBR*_GaN/diamond_ are higher than the ones obtained with SiN (41/30 against 9.5/5.5 m^2^⋅K/GW, respectively [10,82]) and significantly larger than the minimum theoretical value for the GaN/diamond interface (3 m^2^⋅K/GW [82]). During the diamond deposition extensive deterioration of the GaN surface occurs, which results in the appearance of voids measuring up to 50 nm. This results in an extremely rough surface that increases the scattering of the phonons. Smith et al. [76] and Waller et al. [74] reported *TBR*_GaN/diamond_ as high as 220 m^2^⋅K/GW. In both works the GaN surface was seeded using a two-step electrostatic spray technique (see more details in Section 4.1.3.1).

A different possibility is to replace the SiN layer with a layer of material with higher κ, such as AlN [10,82] (30 [46] against 285 W/(m∙K) [165], respectively, in crystalline forms). However, AlN thin films decompose in hydrogen (H_2_) and H_2_/CH_4_ plasmas at low pressures (25–5 Torr) and high temperatures (650–1070 °C) [166], and direct growth of diamond has proven to be very difficult. A possible solution would be the pre-treatment of the AlN surface, prior to exposure to CVD diamond growth conditions. Previous exposure of the AlN substrates to carbon tetrafluoride (CF_4_) plasma allowed increasing the seeding density by nearly 3 orders of magnitude in comparison with untreated substrates [167]. Mandal et al. [84] reported that exposure of the AlN surface to H_2_/nitrogen (N_2_) plasma was necessary for the deposition of thick (>100 μm) and adherent diamond layers. However, the real usefulness of replacing SiN with AlN is doubtful, since AlN and SiN thin films are amorphous and feature similar κ values (≈1–5 nm) [82]. In fact, two different groups reported higher values of *TBR*_GaN/diamond_ obtained with 5 nm-thick AlN layers than with similar SiN layers (18.2/9.5 against 15.9/6.5 m^2^⋅K/GW, respectively [10,82]). In both works this difference was attributed to discontinuities in the AlN layer itself which resulted in the etching of the GaN surface (Figure 21); however it is not clear if the as-deposited AlN layer was discontinuous or if it was etched during the deposition of diamond. Jia et. al [83] observed the same tendency with thicker dielectric layers: the *TBR* obtained with 100 nm-thick AlN layers and SiN layers was 56.4 and 38.5 m^2^⋅K/GW, respectively.

Seeding the pre-treated AlN surfaces with H-terminated detonation nanodiamond (DND) seeds resulted in an average *TBR* of 16 m^2^⋅K/GW. A breakthrough was recently reported by Smith and co-workers [76], who used a two-step electrostatic spray technique to seed 130 nm-thick AlN films deposited on Si substrates. Using this method the experimental *TBR* was as low as 1.47 m^2^⋅K/GW, close to 0.8 m^2^·K/GW, the theoretical minimum *TBR* achievable at the AlN−diamond interface from a diffuse mismatch model, relying only upon the density of states in these two materials [84] (see more details in Section 4.1.3.1).

It should be mentioned that in the experiments by Mandal et al. [84] and Smith et al. [76] the *TBR* values refer to the interface between the diamond and the AlN substrate: as a consequence, no direct comparison can be made with the *TBR*_GaN/diamond_ previously reported by other groups (which includes the interfaces GaN/dielectric layer and dielectric layer/diamond, as well as the *R*_th_ of the dielectric layer itself). Nevertheless, the replacement of the low κ amorphous SiN layer with crystalline AlN is a promising approach. Ideally, the AlN layer should be integrated just below the GaN channel. This layer would act as an etch stop during the device epitaxy, as well as a seed layer for the diamond growth [168]. This is the approach proposed by Field and co-workers [106]; since integrating a thin AlN or high Al content AlGaN layers at this point in the epitaxy is challenging because of alloying with surrounding layers, they used a relatively low Al content crystalline Al_0.32_Ga_0.68_N layer as the etch stop and interlayer and grew diamond following the same procedure as in [84]. Due to the sample layout, a 10 nm-thick SiC layer was formed between the Al_0.32_Ga_0.68_N/diamond interface, which improved the heat transport across the two materials. A *TBR* of 30 m^2^⋅K/GW was measured, a value still much higher than the theoretical minimum of 4 m^2^⋅K/GW obtained for this interface [106].

#### 4.1.3. Optimizing Diamond CVD for Thermal Management Applications

The initial layers of a diamond film typically feature small grains and feature a correspondingly low κ. This effect, combined with the *TBR* at the diamond/dielectric interface, may contribute to an additional *TBR* of 10 m^2^⋅K/GW [79]. The importance of the quality of the diamond nucleation layer was highlighted in a recent work by Song et al. [169], who showed that the *R*_th_ of a 12 finger/30 μm gate pitch GaN-on-diamond HEMT dissipating 5 W/mm would lower from 13.0 to ≈11.0 K⋅mm/W in the absence of phonon scattering by external defects in the GaN layer and interface (a value ≈49% lower than that of a state-of-the-art similar GaN-on-SiC structure). If the κ of the diamond nucleation layer were the same as its bulk conductivity, the *R*_th_ of this devices would further decrease to ≈10.0 K⋅mm/W (≈54% lower than the *R*_th_ of a similar GaN-on-SiC HEMT). Following the same trend, in the case of a HEMT with one finger gate, if the *TBR* at the GaN/diamond interface decreased from 13 to 3 m^2^⋅K/GW, the device *R*_th_ would go from 2.8 to 1.9 K⋅mm/W, which corresponds to a ≈30% reduction in the peak operating temperature at a given power level [73].

The morphology and properties of the diamond nucleation layer are intrinsically related with the process of depositing diamond films on non-diamond substrates. The deposition of diamond on a foreign substrate requires a seeding step, during which the substrate surface is enriched with diamond nanoparticles (DNP). Different techniques can be used for this purpose, such as ultrasonic agitation in a suspension containing DNP or spin-coating of a solution saturated with the same. Once exposed to diamond growth conditions, the diamond seeds grow three-dimensionally and eventually coalesce, forming a closed diamond film. At this stage the individual crystallites start growing perpendicularly to the surface, following the Van der Drift model [170], until growth terminates. The incubation time for the onset of the formation of diamond crystallites can be 15–45 min, depending on the growth parameters.

The growth of the diamond crystals from individual diamond seeds translates in the existence of a so-called diamond nucleation region which contains a high concentration of defects and grain boundaries that increase the phonon scattering and consequently decrease the thermal conductance. The thickness of this nucleation layer ranges typically between 10 and 50 nm, depending on the seeding method and deposition conditions, and its κ can be as low as 3 W/(m∙K) [78]. The appearance of voids at the diamond/substrate interface at the locations where the enlarged diamond seeds touch one another is also common. This effect is represented schematically in Figure 22a [84]. Figure 22b shows a high-angle annular dark-field STEM (HAAD-STEM) image of the interface where such voids can be easily identified.

The size of the diamond grains is typically a few nm close to the substrate and increases with the thickness of the film. The evolution of the grain size has been studied computationally [171,172] and experimentally [173]; it has been shown that, depending on the growth conditions, the lateral size of the grains and their aspect ratio are strongly changing with the film thickness. As a consequence, the grain boundary density varies with the depth of the PCD layer, translating into an inhomogeneous κ_in-plane_. On the other hand, in columnar PCD films κ_out-of-plane_ is typically higher than κ_in-plane_ [174]. However, this condition does not necessarily hold true in the nucleation region, where κ_in-plane_ can be higher than κ_out-of-plane_ and vice-versa for a given PCD film thickness [175]. The grain size dependence of κ, which is especially pronounced near the nucleation region, is therefore a critical parameter for maximizing the heat-spreading capabilities of PCD films on hybrid diamond/GaN devices. The dependence of the in-plane and out-of-plane κ with the PCD film thickness is shown in Figure 23.

##### 4.1.3.1. Impact of the Seeding Procedure

It is obvious that the deposition conditions (CH_4_/H_2_ ratio, substrate temperature, and pressure) impact directly the growth rate as well as the macro-level characteristics/quality of the diamond deposits. Similarly, the seeding step has a considerable impact on the quality of the diamond film in the nucleation region, close to the interface with the substrate.

Increasing the seeding density has been the motto of researchers during the past years. Given their small size, which allows for homogeneous and high density seeding, DND seeds are the most frequently used diamond particles. The seeding methods that lead to the highest seeding densities include the ultrasonic agitation in a suspension containing DND seeds and the enhancement of the electrostatic attraction between the DND seeds and the substrate.

Ultrasonic agitation in a suspension containing DND seeds. Ultrasonic seeding has been widely used since the early 1990s [176]. Seeding can be performed with different sized diamond grit, as well as with a mixture of diamond grit and metal particles. As an example, the adhesion of 0.25 µm tungsten (W) and Ti particles to a nanodiamond suspension was reported to increase the seeding density and the adhesion of diamond films deposited on Si and Si_3_N_4_ substrates [177]. A significant improvement was achieved with DND particles colloidal solutions, which enabled seeding densities in excess of 10^12^ cm^−2^ [178].Enhancement of electrostatic attraction between DND seeds and substrate. By properly tuning the ζ-potentials of seeds and/or substrates, one can benefit from enhanced electrostatic attraction between the DND seeds and the substrate. This can be achieved by terminating the DND seeds with oxygen (O) or H atoms [179] or by preparing polymer/nanodiamond colloids [180,181]. This effect can be further enhanced by exposing the substrate to a plasma treatment in order to guarantee that the ζ-potentials of diamond seeds and substrate have opposite signs [84].

Despite leading to the highest reported seeding densities, DND particles may not be the best choice for thermal management applications, which rely on the minimization of the *TBR* between diamond and substrate. DND particles possess an amorphous shell [182], which may contribute to the increase of sp^2^ bonds close to the interface, which in turn will enhance phonon scattering, thus compromising the thermal transport across the interface. In addition, while it is true that a high seeding density guarantees a lower coalescence time, it is not absolutely clear that this is an advantage for thermal applications. A very high density of seeds means that they are not allowed to grow significantly before they coalesce with each other; as a consequence the amount of defects and grain boundaries further increases—and so does the scattering of the phonons.

If seeding is performed with larger sized particles, the grain/grain boundaries ratio might be maximized. Following this reasoning, in 2017 Liu and co-workers evaluated the effect of seeding GaN substrates with a 30 nm-thick SiN protective layer with 30 and 100 nm diamond seeds [81]. Unlike what happened with the smaller 30 nm particles, seeding with 100 nm particles damaged the SiN layer, which resulted in the etching of the GaN surface and appearance of pin holes during diamond growth. More recently, Bai et al. [183] evaluated the impact of seeding with 4 and 20 nm seeds on the κ of diamond films deposited in Si substrates. Seeding with the larger 20 nm seeds resulted in a smaller seeding density (7 × 10^9^ cm^−2^ in contrast with 3 × 10^11^ cm^−2^ with the 4 nm seeds) but resulted in larger-sized grains near the interface region, and in a correspondingly higher in-plane κ as measured by Raman thermography.

In order to overcome the limitations of the standard seeding procedures, Smith et al. [76] proposed a two-step electrostatic spray technique to seed 130 nm-thick AlN films deposited on Si substrates. Using this method, the surface of the AlN was initially seeded with 2 µm diamond particles with smooth facets, which guaranteed a large contact area with the AlN surface and favored the thermal transport across the AlN/diamond interface. Following this step, the substrate was electrostatically sprayed again with 3.3 nm DNP, which filled in the gaps between the larger seeds and prevent the formation of voids. The schematic diagram explaining the rationale for the two-step seeding is shown in Figure 24. The advantages of this method are twofold: the microparticles of diamond guarantee a lower grain boundary ratio (when compared to conventional seeding with DND particles) and the DNP fill the voids between the larger particles, protecting the AlN surface from the plasma. This layer of electrosprayed seeds replaces the highly defective diamond nucleation layer characteristic of heteroepitaxial diamond films. Using this method, an extremely low *TBR* of 1.47 m^2^⋅K/GW (close to the theoretical minimum of 0.8 m^2^⋅K/GW [84]) was obtained at a diamond/AlN interface.

#### 4.1.4. Optimizing the Thickness of the GaN Epilayers

The determination of the GaN buffer layer thickness typically takes into account the electrical performance of the device and material quality requirements, instead of thermal requirements [184]. In latest generation GaN-on-diamond devices the low thermally conductive nucleation and strain relief layers are removed before the deposition of the diamond film, and as a consequence the thickness of the GaN buffer layer has a non-negligible impact on the total *R*_th_ of the device. In fact, in a joint work between the University of Bristol and Element Six [88], the thickness of the GaN buffer layer was reduced from 700 to 354 nm and the thickness of the dielectric layer to 17 nm, leading to a device *R*_th_ of 9 K⋅mm/W, a value significantly lower than the *R*_th_ of GaN-on-SiC devices (16 K⋅mm/W). Besides lowering the maximum temperature, the thinning of the GaN layer decreased self-heating, resulting in a smaller change in output conductance and providing a means to reduce the thermally-generated device non-linearities.

Taking into account these results, it might initially be assumed that the GaN layer should be as thin as possible. If it is too thick, the *R*_th_ associated with the GaN layer and consequently the *R*_th_ of the device increase. However, if the GaN layer is too thin (especially when the heat source length is comparable to the device length and for small *TBR*_GaN/diamond_ values), the concentrated heat flux coming out from the heat source reaches the GaN/diamond interface without spreading, causing the region right under the heat source to heat up significantly and leading to an increased *R*_th_ [85]. As an example, the peak channel temperature of a 4 × 125 µm/40 µm gate pitch GaN-on-diamond HEMT dissipating 10 W/mm is shown in Figure 25 for different values of *TBR*_GaN/diamond_ and GaN layer thicknesses (from [185]). It can be seen that thinner GaN layers may lead to lower or higher peak channel temperatures, depending on the particular value of the *TBR*_GaN/diamond_.

Simulations have systematically shown that the device *R*_th_ monotonically decreases with increasing GaN layer thickness until it reaches a minimum. The higher the κ of the diamond, the lower the *TBR*_GaN/diamond_ [86,87,89], and the smaller the hotspot area [85], the more important this dependence becomes.

In 2020, Song and co-workers [90] have shown that, while the device *R*_th_ is fairly low at a typical GaN thickness of 1 μm (≈12.9 and ≈16.4 K⋅mm/W for *TBR*_GaN/diamond_ of 6.5 and 30 m^2^⋅K/GW, respectively), a reduction in the GaN thickness below 1 μm may result in a substantial increase in the device *R*_th_, in particular when *TBR*_GaN/diamond_ is high (≈31% and ≈118% increase for *TBR*_GaN/diamond_ of 6.5 and 30 m^2^⋅K/GW, respectively, and 0.1 µm GaN thickness). For the same *TBR*_GaN/diamond_ values, the GaN thickness that minimizes the *R*_th_ of the device is ≈3.6 and ≈5.8 μm, respectively, and the minimum *R*_th_ is ≈5% and ≈19% the *R*_th_ with 1 µm of GaN.

From what was presented above, it can easily be concluded that in order to take the most benefit of the diamond substrate, the impact of the GaN buffer layer thickness on the overall *R*_th_ of the device should also be taken into account in the design phase, in addition to the traditionally considered electrical performance and material quality requirements.

### 4.2. Challenges of Bonding GaN and Diamond Wafers

SAB and VdW bonding are the most promising methods for bonding diamond GaN and diamond substrates. Both methods have a few advantages in comparison to the fabrication of GaN-on-diamond wafers by direct diamond CVD. To begin with, the GaN wafers without the nucleation and strain relief layers can be bonded to SCD plates that feature an extremely high κ, whereas the κ of the films close to the GaN on the GaN-on-diamond wafers is quite low. In addition, only an extremely thin Si layer (or no layer, in the case of VdW bonding) is required to bond the two materials, which allows the *TBR*_GaN/diamond_ to be significantly reduced. Finally, the bonding is performed at a temperature significantly lower than the 700–800 °C required to deposit good quality PCD (RT in the case of SAB, 300 °C in the case of VdW bonding), which means there is no residual stress (or that it is minimal) at the diamond/GaN interface.

But the bonding of the wafers also presents some drawbacks. The area of available SCD substrates is quite small (only a few mm^2^), which seriously compromises the scalability of the process. Bonding GaN wafers with large area PCD substrates could in principle overcome this limitation; however, bonding with PCD substrates is not reproducible in the case of VdW bonding. In a recent work, the SAB of GaN and PCD with a Mo/Au interlayer has been reported and the mechanical strength of the interface was evaluated, but the thermal characterization is still lacking.

Despite these limitations, the potential of the technique has been recognized by Fujitsu and Mitsubishi, and it is expected that commercial devices will be available in a near future. The SAB of GaN HEMTs and large area PCD substrates is also regarded as an area worth of investigation, since the successful bonding of GaN and PCD would allow the fabrication of large area GaN/diamond wafers with improved heat extraction.

### 4.3. Challenges of Epitaxially Growing GaN on Diamond 

The GaN epitaxy allows the deposition of the GaN layers directly onto high thermally conductive diamond substrates without the need of any dielectric layer. However, the required AlN nucleation and stress-relief AlGaN/GaN stacked layers with lower κ will themselves hamper the flow of the heat from the top HEMT structures to the back of the diamond substrates. In addition, if the epitaxy is performed on SCD substrates, the area will be limited to a few mm^2^. This means that the true benefit of GaN epitaxy is doubtful when compared with the direct bonding of the HEMT structures and the diamond substrates.

A breakthrough has been achieved with the ELO of the GaN layers on PCD substrates, thus overcoming the small area availability of the SCD substrates. Nevertheless, if the GaN is deposited directly on the PCD substrate, as proposed by Webster et al. [124], the low thermally conductive stress-relief layers will once again compromise the flow of heat. In the approach followed by Ahmed et al. [137], on the other side, the GaN is initially deposited on a Si substrate and selectively deposited PCD stripes replace the SiN stripes used in [124] for the ELO step. The final integration of the ELO grown GaN epilayers with the thick diamond heat spreaders will require a few more steps: (i) the MOCVD of the HEMT structures, (ii) the removal of the Si substrate and AlGaN/GaN stress-relief layers, and (iii) the direct CVD of the diamond film. While (ii) allows placing the diamond directly in contact with the GaN epilayers, (iii) will require the deposition of a dielectric layer on the exposed GaN, as in the case of direct diamond CVD. The structure of the final GaN-on-diamond wafers fabricated using this method will be similar to the structure of GaN-on-diamond wafers described in Section 3.1, but with a difference, since the diamond stripes embedded in the ELO GaN can be overgrown by the thick diamond without the need of a protective dielectric layer. Despite this improvement, the evaluation of the potential of this approach needs to take into consideration that the fabrication procedure is significantly more complex than that of standard GaN-on-diamond wafers.

### 4.4. Challenges of Capping GaN HEMTs with Diamond

The capping of the HEMT devices with a diamond film is the technically simplest approach. In theory, the GaN HEMTs can be capped with the diamond film after the passivation steps without any changes in the fabrication procedure. However, due to the nature of the diamond CVD process, the diamond-capped HEMTs face some of the same issues as the GaN-on-diamond wafers.

On one side, the passivation layer, though preventing the degradation of III-nitride layers in the harsh CVD environment, contributes with a non-negligible *TBR* which hampers the flow of heat towards the diamond heat spreader. On the other side, the quality—and hence the κ—of the diamond film close to the interface also play a critical role. In this sense, the discussion presented in Section 4.1.2 and Section 4.1.3 holds valid for the capping diamond approach. In addition, the top AlGaN barrier layer features a low κ (Table 2) and introduces a non-negligible *R*_th_ between the hot spot and the capping diamond layer.

Finally, thermal stresses will inevitably accumulate at the GaN/diamond interface due to the *CTE*s mismatch. Any change in the stress−strain state in AlGaN/GaN heterostructures, especially in the fully strained pseudomorphically grown AlGaN barrier layer, would have a significant impact on the 2DEG characteristics. Therefore, the proper understanding of the impact the stress-strain state induced by the diamond CVD on the 2DEG characteristics will require a thorough evaluation of the barrier layer stress.

Nevertheless, these limitations did not prevent Fujitsu from developing diamond capped GaN HEMTs with improved heat dissipating capabilities, meaning that capping HEMTs with diamond may provide a valuable way of improving the thermal management of these devices.

### 4.5. What Is the Best Approach?

From what was described in the previous sections, it can be concluded that there is no universally best approach, and each one has its own pros and cons.

GaN-on-diamond is undoubtedly the more mature technology. Large-area deposition capability is a significant advantage and, despite being relatively complex, the fabrication process has been optimized and the main issues have, at least partially, been solved. Other companies, in addition to Element Six, have been involved in the process. In 2019 RFHIC reported a manufacturing procedure that allows the fabrication of 4″ GaN-on-diamond HEMTs using a laser drilling process [102]. GaN-on-diamond HEMTs and RF power amplifiers can be currently purchased from Qorvo and Akash Systems, Inc. However, and despite the maturity and success of this technology, some room for improvement still exists (namely the decrease of the gate leakage current), and exciting improvements are expected in the coming years.

The bonding of GaN and diamond SCD substrates is also reaching a high level of maturity, and companies such as Mitsubishi Electric Corp. and Fujitsu Limited have reported the successful transfer of GaN and GaN-on-SiC HEMTS to diamond substrates. This anticipates a bright future for bonded GaN/diamond devices. On one side, these devices show a potential for decreasing the *TBR*_GaN/diamond_ below the minimum achievable with GaN-on-diamond technology, since the 30 nm-thick SiN dielectric layer can be replaced by a 2–10 nm-thick Si-based layer. However, since the reported GaN HEMTs have been bonded to SCD substrates, this technique will have a significantly lower yield than the GaN-on-diamond approach and the full scaling up of the technology will be more challenging. If, however, future research deems the bonding of GaN HEMTs and large area PCD substrates feasible and reproducible this technique may compete with the GaN-on-diamond technology.

The epitaxy of GaN on diamond substrates, though feasible, may not bring any realistic advantage. On one side, the best quality GaN films have been grown on low-area SCD substrates. The ELO of GaN has made growth of GaN films with low dislocation density possible on PCD substrates, however functional HEMTs are yet to be demonstrated. On the other side, in the majority of approaches reported so far the nucleation and strain relief layers are part of the final HEMT material stack, and they will hinder the transport of heat to the diamond substrate.

Despite the low κ of the AlGaN barrier layer the capping of passivated HEMTs with a thin diamond film is expected to decrease the peak temperature of the devices between 8% and 20% in comparison with GaN-on-SiC and GaN-on-Si HEMTs, respectively. Despite being a relatively modest number—if compared with the improvement obtained with latest generation GaN-on-diamond devices—this approach is technologically simple, since the diamond can be deposited directly on the passivation layer, and allows for large area growth. A bright future is anticipated also for this approach, as proven by the work recently reported by Fujitsu researchers. 

The comparative advantages and disadvantages of each approach are further summarized in Table 7.

## 5. Conclusions

The integration of diamond and GaN devices has been an active research topic for 20 years. The involvement of companies like Fujitsu and Mitsubishi, for instance, is representative of the impact that hybrid GaN/diamond electronic devices can have on some applications.

The integration of diamond and GaN has been achieved by different methods: the direct CVD of the diamond films on the back of GaN wafers, the bonding of HEMTs and diamond substrates, the direct epitaxy of the GaN layers on diamond substrates, and the diamond capping of passivated HEMTs. The technological advances, the room for improvement, and the advantages/disadvantages of each method have been presented and discussed.

Generally speaking, the fabrication of diamond-on-GaN wafers by direct diamond CVD on the back of the GaN wafers has been quite successful and commercial RF power amplifiers fabricated on GaN-on-diamond wafers are currently available for satellite communications. The bonding of GaN HEMTs and SCD substrates and the capping of GaN HEMTs have also been raising interest from companies such as Mitsubishi Electric Corp. and Fujitsu Limited. Recent advances in the epitaxial growth of GaN layers on PCD substrates anticipate interesting technological developments in a near future. Far from having reached the limits of the technology, it can be thus said that the integration of diamond and GaN will remain an active research topic in the years to come, involving academic and industrial players, with the ultimate goal of increasing the power density and reliability of GaN HEMTs. 

## Figures and Tables

**Figure 1 materials-15-00415-f001:**
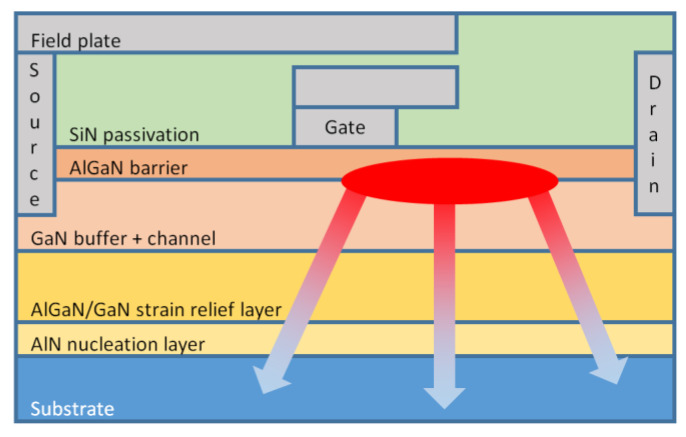
Simplified representation of an AlGaN/ GaN HEMT structure; drawing not to scale.

**Figure 2 materials-15-00415-f002:**
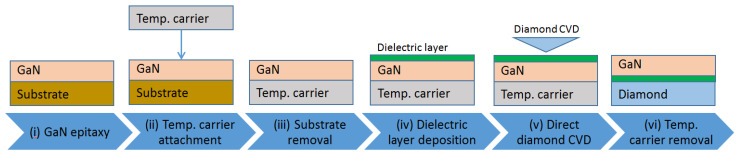
GaN-on-diamond fabrication steps.

**Figure 3 materials-15-00415-f003:**
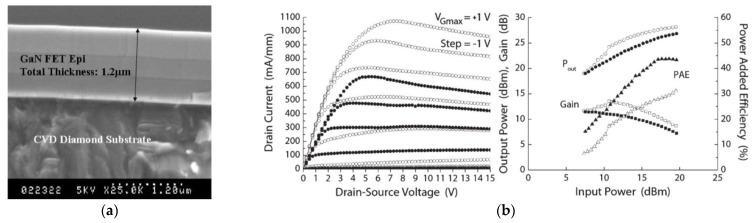
(**a**) SEM cross-section of a GaN-on-diamond wafer (© 2006 IEEE. Reprinted, with permission, from G. H. Jessen et al., “AlGaN/GaN HEMT on diamond technology demonstration,” Tech. Dig.—IEEE Compd. Semicond. Integr. Circuit Symp. CSIC, pp. 271–274, 2006 [63]). (**b**) Electrical characteristics of similar GaN-on-diamond (solid symbols) and GaN-on-SiC (open symbols) HEMTs; output power and *PAE* measured at 10 GHz CW class B operation and 20 V *V*_D_ (© 2007 IEEE. Reprinted, with permission, from J. G. Felbinger et al., “Comparison of GaN HEMTs on diamond and SiC substrates,” IEEE Electron Device Lett. [65]).

**Figure 4 materials-15-00415-f004:**
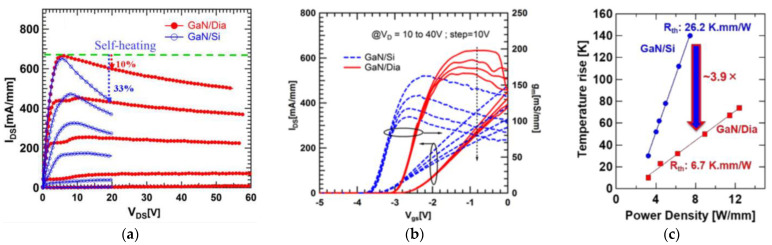
(**a**) *I*_D_ versus *V*_D_ characteristics, (**b**) Transfer characteristics for different *V*_D_, and (**c**) Temperature rise versus *P*_D_ for similar GaN-on-diamond and GaN-on-Si HEMTs (reprinted from [28]; permission conveyed through CCBY 4.0: https://creativecommons.org/licenses/by/4.0/~ (accessed on 7 October 2021)).

**Figure 5 materials-15-00415-f005:**
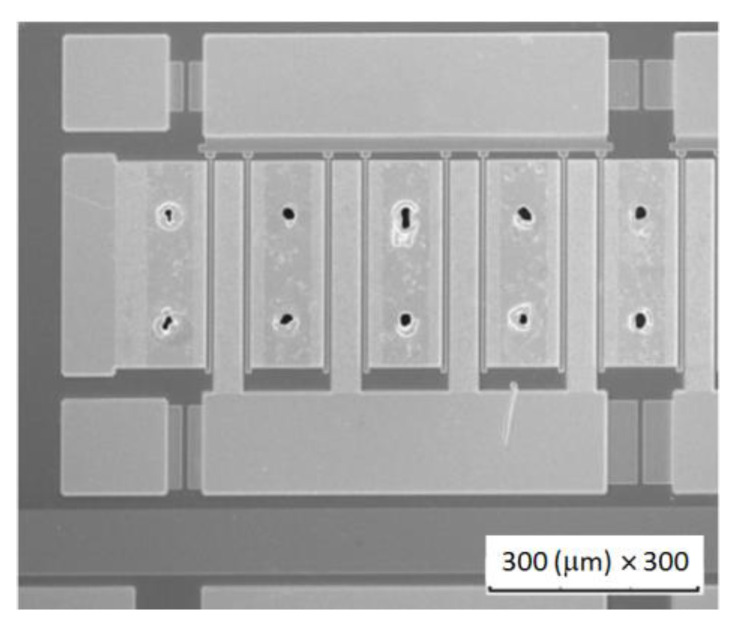
Inner slot via hole shape on source pad (© 2019 IEEE. Reprinted, with permission, from W. S. Lee, et al., “A GaN/Diamond HEMTs with 23 W/mm for Next Generation High Power RF Application,” in IEEE MTT-S International Microwave Symposium Digest, 2019, vol. 2019-June, pp. 1395–1398 [102]).

**Figure 6 materials-15-00415-f006:**
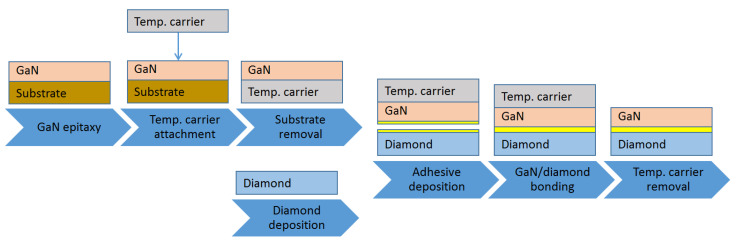
Bonding of GaN and diamond substrates.

**Figure 7 materials-15-00415-f007:**
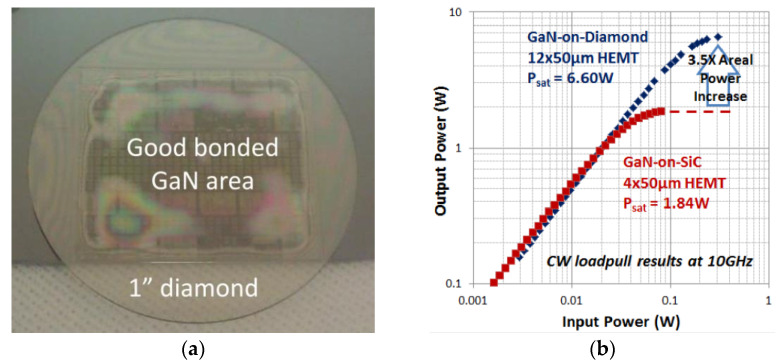
(**a**) GaN HEMTs bonded to a 1″ PCD substrate (© 2014 IEEE. Reprinted, with permission, from K. K. Chu et al., “S2-T4: Low-temperature substrate bonding technology for high power GaN-on-diamond”, Lester Eastman Conf. 2014—High Perform. Devices, LEC 2014, pp. 1–4, 2014 [109]). (**b**) Comparison of input-output power curves obtained at 10 GHz for GaN/diamond HEMT with 3 times larger gate periphery than GaN-on-SiC HEMT (© 2015 IEEE. Reprinted, with permission, from K. K. Chu et al., “High-Performance GaN-on-Diamond HEMTs Fabricated by Low-Temperature Device Transfer Process,” 2015 IEEE Compd. Semicond. Integr. Circuit Symp. CSICS 2015, pp. 7–10, 2015 [111]).

**Figure 8 materials-15-00415-f008:**
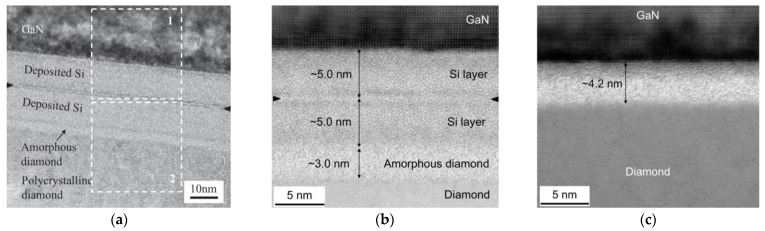
Cross-section TEM image of GaN/diamond interface obtained by SAB at room temperature after (**a**) sputtering a ≈12 nm (Reprinted from Scr. Mater., vol. 150, F. Mu et al., "Room temperature GaN-diamond bonding for high-power GaN-on-diamond devices", pp. 148–151 [113], Copyright 2018, with permission from Elsevier) and (**b**) ≈5 nm (Reprinted with permission from Z. Cheng et al., “Interfacial Thermal Conductance across Room-Temperature Bonded GaN-Diamond Interfaces for GaN-on-Diamond Devices,” ACS Appl. Mater. Interfaces, vol. 12, pp. 8376–8384, 2020 [115]. Copyright 2020 American Chemical Society) Si nanolayer on both surfaces; (**c**) activating both surfaces with a mixed beam of Si and Ar ions (Reprinted with permission from Z. Cheng et al., “Interfacial Thermal Conductance across Room-Temperature Bonded GaN-Diamond Interfaces for GaN-on-Diamond Devices,” ACS Appl. Mater. Interfaces, vol. 12, pp. 8376–8384, 2020 [115]. Copyright 2020 American Chemical Society).

**Figure 9 materials-15-00415-f009:**
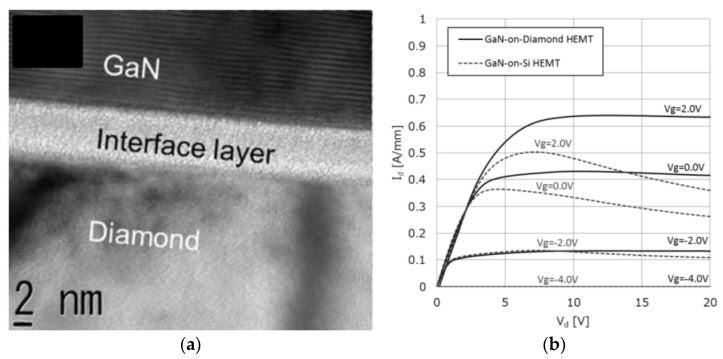
(**a**) Cross-section TEM image of GaN/diamond interface obtained by SAB. (**b**) *I*_D_-*V*_D_ characteristics of GaN-on-Si (dashed lines) and GaN/diamond (solid lines) HEMTs (reprinted from [114], Copyright 2019 The Japan Society of Applied Physics).

**Figure 10 materials-15-00415-f010:**
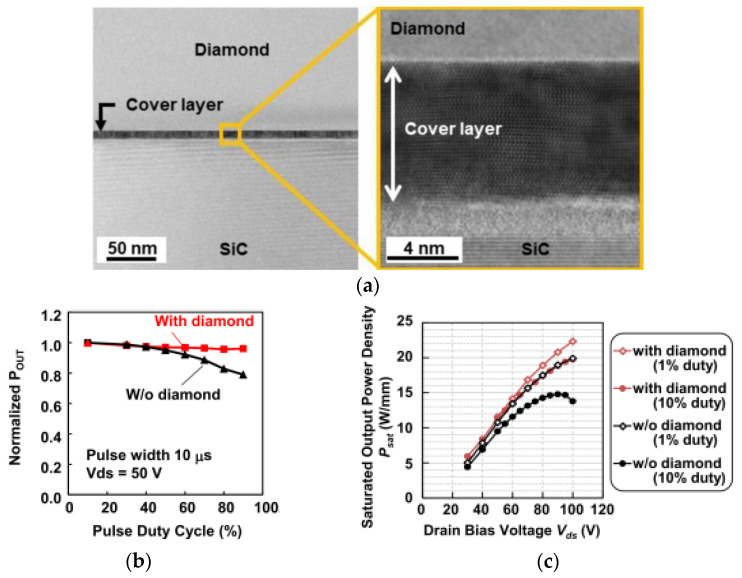
(**a**) Cross-section TEM image of SiC/SCD interface obtained by SAB after protecting the diamond surface with a Ti film. (**b**) Normalized *P*_D_ vs. duty-cycle. (**c**) *P*_D_ as a function of *V*_D_ (reprinted from [118], Copyright 2020 The Japan Society of Applied Physics).

**Figure 11 materials-15-00415-f011:**
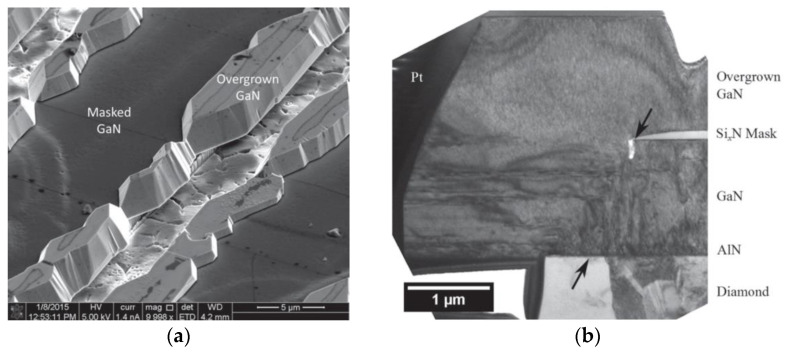
(**a**) SEM image of masked and ELO growth of GaN. (**b**) TEM image showing original and ELO growth of GaN; arrows indicate boundary between original (right of boundary) and ELO (left of boundary) growth (reprinted from [124]; permission conveyed through CCBY 3.0: https://creativecommons.org/licenses/by/3.0/ (accessed on 7 October 2021)).

**Figure 12 materials-15-00415-f012:**
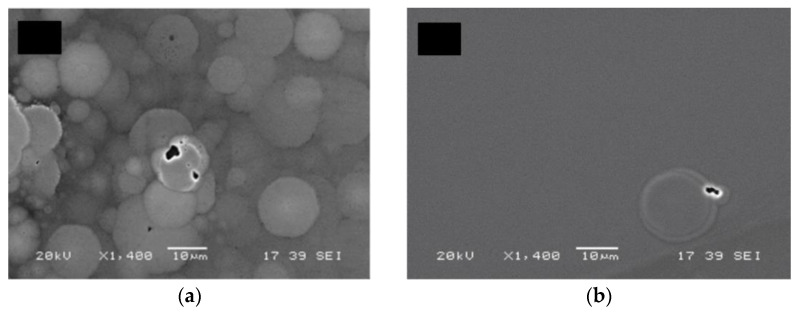
SEM images showing improvement in surface morphology of thicker GaN films. (**a**) 300 nm- and (**b**) 1.1 µm-thick GaN films (republished with permission of IOP Publishing, from “Growth of GaN epitaxial films on polycrystalline diamond by metal-organic vapor phase epitaxy," Q. Jiang, D. W. E. Allsopp, and C. R. Bowen, vol. 50, no. 16, 2017 [136]; permission conveyed through Copyright Clearance Center, Inc.).

**Figure 13 materials-15-00415-f013:**
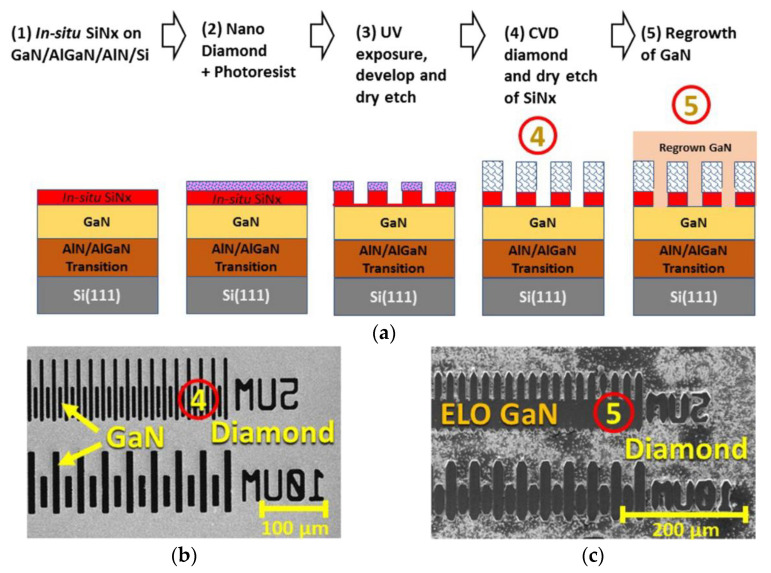
(**a**) Processing steps of epitaxial growth of GaN on PCD. SEM images of (**b**) selectively deposited PCD stripes (after step 4) and (**c**) same features after ELO of GaN (after step 5) (reprinted with permission from R. Ahmed et al., “Integration of GaN and Diamond Using Epitaxial Lateral Overgrowth,” ACS Appl. Mater. Interfaces, vol. 12, no. 35, pp. 39397–39404, 2020 [137]. Copyright 2020 American Chemical Society).

**Figure 14 materials-15-00415-f014:**
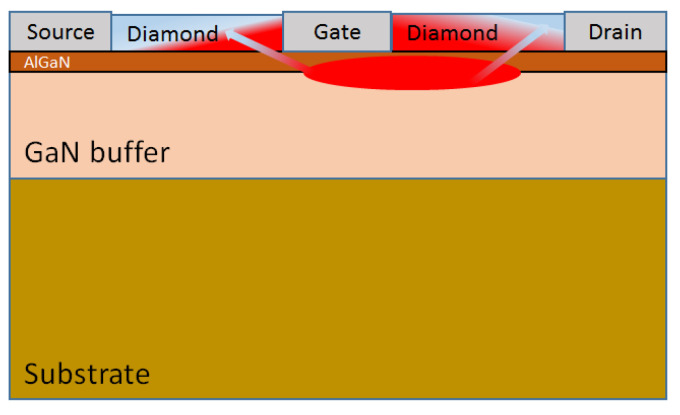
Capping diamond concept.

**Figure 15 materials-15-00415-f015:**
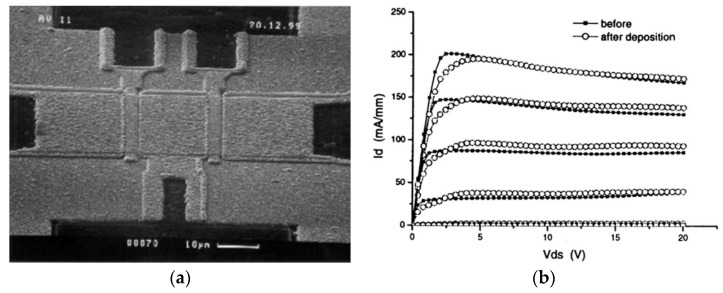
(**a**) Reflection electron microscope image (REM) and (**b**) Transfer characteristics of a two-finger GaN FET coated with a 0.7 µm-thick PCD film (reprinted from Diam. Relat. Mat., vol. 10, no. 3–7, M. Seelmann-Eggebert, et al., “Heat-spreading diamond films for GaN-based high-power transistor devices,” pp. 744–749 [46], Copyright 2001, with permission from Elsevier).

**Figure 16 materials-15-00415-f016:**
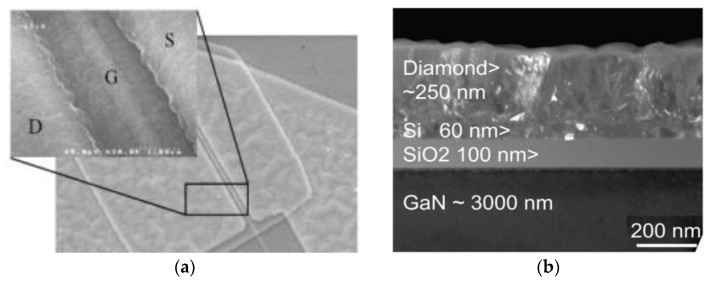
(**a**) SEM top view of a GaN HEMT coated with a 0.5 µm-thick NCD layer. (**b**) TEM cross section of the NCD film showing no voids at the passivation/growth interface (reprinted from Diam. Relat. Mater., vol. 20, no. 4, M. Alomari et al., “Diamond overgrown InAlN/GaN HEMT,” pp. 604–608 [148], Copyright 2011, with permission from Elsevier).

**Figure 17 materials-15-00415-f017:**
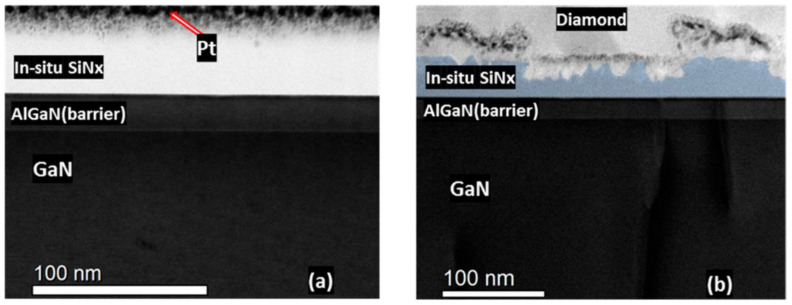
Bright-field STEM images showing the Al(Ga)N/SiN layer interfacial region (**a**) before and (**b**) after diamond growth (reprinted with permission from A. Siddique et al., “Structure and Interface Analysis of Diamond on an AlGaN/GaN HEMT Utilizing an in Situ SiNx Interlayer Grown by MOCVD,” ACS Appl. Electron. Mater., vol. 1, pp. 1387–1399, 2019 [91]. Copyright 2019 American Chemical Society).

**Figure 18 materials-15-00415-f018:**
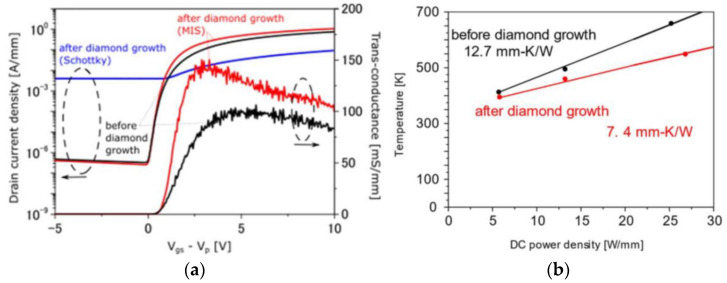
(**a**) Transfer curve and transconductance vs. gate voltage. (**b**) Hotspot temperature calculated from the Raman peak shift (reprinted from [154], Copyright 2021 The Japan Society of Applied Physics.

**Figure 19 materials-15-00415-f019:**
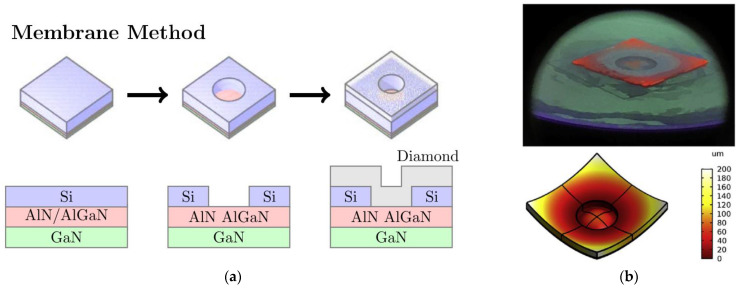
(**a**) Fabrication steps of a GaN-on-diamond membrane. (**b**) Image of the 5 mm-diameter membrane inside the MPCVD system and corresponding thermally-induced mechanical displacement (reprinted from [95]; permission conveyed through CCBY 4.0: https://creativecommons.org/licenses/by/4.0/ (accessed on 7 October 2021)).

**Figure 20 materials-15-00415-f020:**
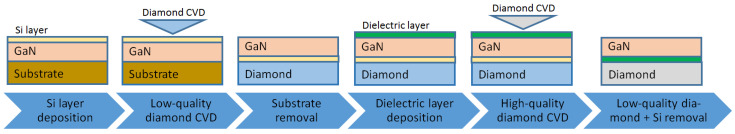
Fabrication of GaN-on-diamond wafers by dual-sided diamond deposition.

**Figure 21 materials-15-00415-f021:**
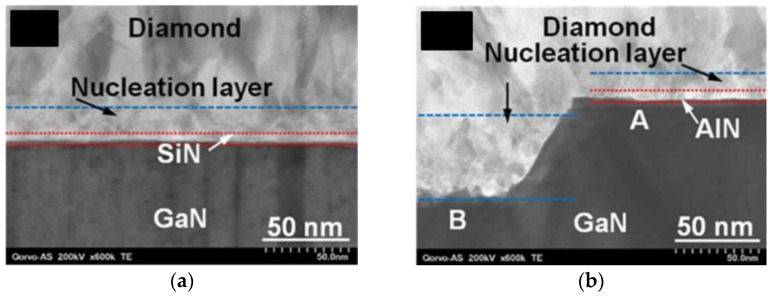
TEM cross-sections of GaN/diamond interfaces with a 5 nm-thick barrier layer of (**a**) SiN and (**b**) AlN. Uniform and smooth GaN/dielectric and dielectric/diamond interfaces are obtained with the SiN layer. With AlN, some regions (A) show smooth interfaces, however in other regions (B) ≈60 nm of GaN has been etched away (reprinted with permission from Y. Zhou et al., “Barrier layer optimization for enhanced GaN-on-diamond device cooling,” ACS Appl. Mater. Interfaces, vol. 9, no. 39, pp. 34416–34422, 2017 [82]. Copyright 2017 American Chemical Society).

**Figure 22 materials-15-00415-f022:**
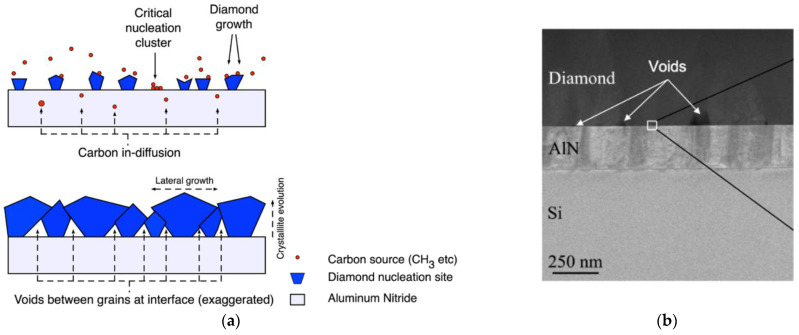
(**a**) Schematic of diamond film growth with low nucleation density; (**b**) HAAD-STEM image of the interface where the voids are clearly seen (reprinted from [84]; permission conveyed through CCBY 4.0: https://creativecommons.org/licenses/by/4.0/ (accessed on 7 October 2021)).

**Figure 23 materials-15-00415-f023:**
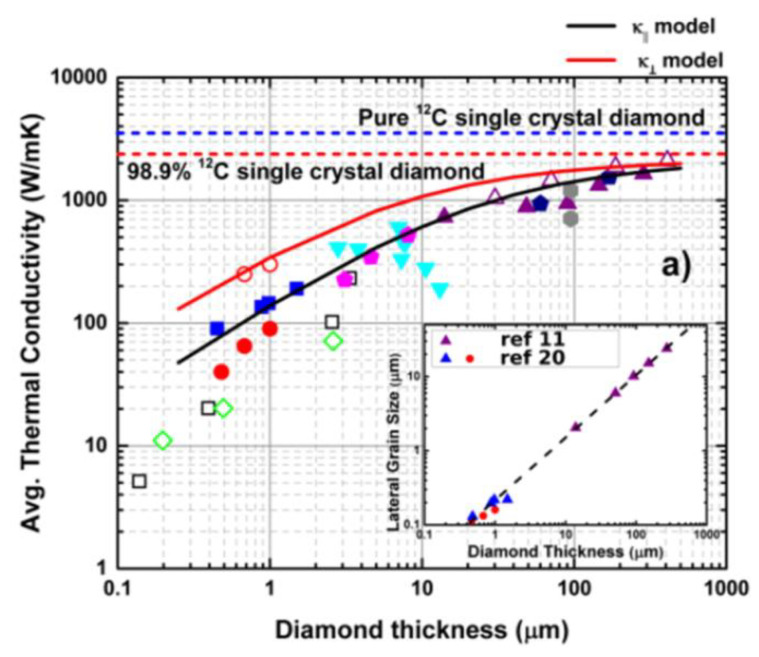
Evolution of in-plane and out-of-plane κ with diamond film thickness (© 2016 IEEE. Reprinted, with permission, from J. Anaya, et al., “Thermal management of GaN-on-diamond high electron mobility transistors: Effect of the nanostructure in the diamond near nucleation region,” in 15th IEEE Intersociety Conference on Thermal and Thermomechanical Phenomena in Electronic Systems (ITherm), 2016, pp. 1558 1–8 [175]).

**Figure 24 materials-15-00415-f024:**
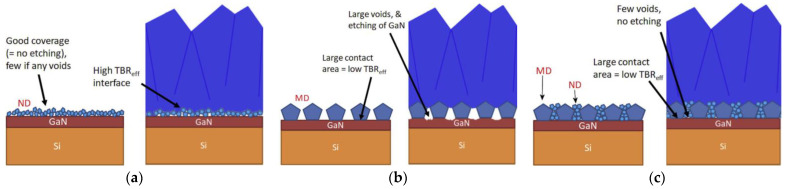
Rationale behind the two-step seeding: seeding with (**a**) ND particles alone, (**b**) diamond microparticles alone, and (**c**) diamond microparticles followed by ND (reprinted from Carbon, vol 167, E. J. W. Smith et al., “Mixed-size diamond seeding for low-thermal-barrier growth of CVD diamond onto GaN and AlN”, pp. 620–626 [76], Copyright 2020, with permission from Elsevier).

**Figure 25 materials-15-00415-f025:**
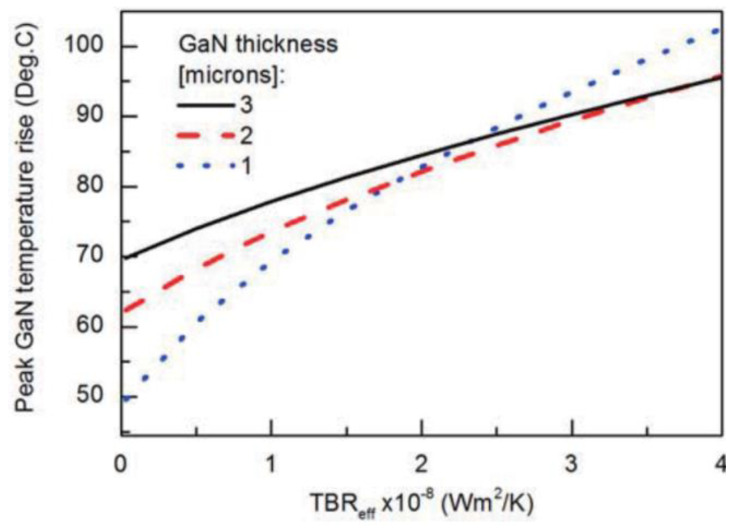
Peak channel temperature of a 4 × 125 µm/40 µm gate pitch GaN-on-diamond HEMT dissipating 10 W/mm for different *TBR*_GaN/diamond_ values and GaN layer thicknesses (Copyright © 2014 IEICE. Reprinted, with permission, from M. Kuball, J. A. Calvo, R. B. Simon, and J. W. Pomeroy, “Novel thermal management and its analysis in GaN electronics,” Asia-Pacific Microw. Conf., pp. 920–922, 2014 [185]).

**Table 1 materials-15-00415-t001:** Selected properties of relevant semiconductors [1,6,7,8,9,10,11,12,13].

Material Property	Si	SiC	GaN	Diamond
Bandgap (eV)	1.12	2.9 (6H)/ 3.2 (4H)	3.4	5.47
Breakdown field (×10^6^ V/cm)	0.25	2.5 (6H)/ 3 (4H)	3–3.75	20
Electron mobility (cm^2^/(V⋅s))	1350	415 (6H)/ 950 (4H)	1100–1300	2400
Thermal conductivity (W/(m⋅K))	150	380–450	130–210	2200 (single crystal) >1800 (polycrystalline)
Coefficient of thermal expansion (×10^−6^ K^−1^)	2.6	3.08	5.6 (*a*_0_)/ 3.2 (*c*_0_)	0.8

**Table 2 materials-15-00415-t002:** Typical thermal conductivity of the materials present in an AlGaN/GaN HEMT [1,7,9,10,11,46,47,48].

Material	Thermal Conductivity (W/(m⋅K))
Sapphire	46
Si	150
SiC	380–450
Single-crystal diamond	2200
Polycrystalline diamond	>1800
GaN	130–210
AlN	130
Al_x_Ga_1−x_N	10–11
In_x_Ga_1−x_N	5–11
SiN	1.6

**Table 3 materials-15-00415-t003:** Evolution of GaN-on-diamond technology.

Ref.	Year	Dielectric Layer		Diamond Film		*TBR*_GaN/diamond_ (m^2^⋅K/GW)
		Thickness (nm)	Material	CVD Type/Thickness (µm)	κ (W/(m⋅K))	
[63]	2006	--	--	HFCVD/25	--	--
[66]	2009	--	--	75	--	--
[67]	2010	--	--	MPCVD/100	>1500	--
[69]	2013	40	--	30	--	36 ± 12 ^b^
[70]	2013	50	--	HFCVD/100	--	18 ^a^
[103]	2014	30	--	100	--	29 ± 2 ^b^
[72]	2014	25	--	HFCVD/95	710	27 ± 3 ^a^
		50		MPCVD/120	1200	36 ^a^
[77]	2014	50	--	MPCVD/110	1200	17 ^b^
		90				41 ^b^
[104]	2014	--	--	MPCVD	1600	19 ± 3 ^b^
[105]	2014	--	--	100	--	47.6 ^b^
						19 ^b^
[78]	2015	34	SiN	HFCVD/100	650	25 ± 3 ^b^
		100		MPCVD/100	1500	50 ± 5 ^b^
		28		MPCVD/100	1500	12 ± 2 ^b^
[79]	2016	40	SiN	MPCVD/100	1370	25.5 ± 0.5 ^b^
						31.0 ± 0.7 ^b^
[80]	2017	31	SiN	HFCVD/100	--	31.8 ± 5.3 ^b^
		22	SiN	MPCVD/100		19.8 ± 4.1 ^b^
						17.4 ± 3.0 ^b^
[81]	2017	30	SiN	MPCVD/100	--	23 ± 3 ^b^
[82]	2017	5	SiN	MPCVD/1	100–700	6.5 ^b^
			AlN			15.9 ^b^
		No interlayer				61.1 ^b^
[10]	2018	5	SiN	MPCVD/1	--	9.5 + 3.8/−1.7 ^b^
			AlN			18.2 + 1.5/−3.6 ^b^
		No interlayer				41.4 + 14.0/−12.3 ^b^
[83]	2019	100	SiN	MPCVD/2	--	38.5 ± 2.4 ^b^
		100	AlN			56.4 ± 5.5 ^b^
[102]	2019	35	SiN	MPCVD/120	--	31.0 ± 3.1 ^b^
[99]	2019	50	SiN	MPCVD/100	--	33
		36	SiN			22
		41	SiN			15
[88]	2019	36	SiN	MPCVD/75	--	20 ^c^
		17				13 ^c^
[101]	2019	30	SiN	MPCVD/100	--	18 ^b^
[106]	2020	20	Al_0.32_Ga_0.68_N	MPCVD/35	--	30 ± 5 ^b^

Values obtained with ^a^ Raman thermography, ^b^ transient thermoreflectance, and ^c^ from luminescence spectra.

**Table 4 materials-15-00415-t004:** Evolution of GaN/diamond bonded wafers technology.

Ref.	Year	Adhesive Layer		Diamond Substrate	Bonding Process	*TBR* (m^2^⋅K/GW)
		Thickness (nm)	Material			
[107]	2013	--	Si-based	1″ × 1″ PCD	Pressing at RT	--
[109]	2014	35	Si-based	1″ PCD wafer	Pressing <150 °C	34 ± 5 ^a^
[112]	2017	30–40	--	3″ PCD wafer	Pressing 180 °C	51 ^b^
[113]	2018	24	Si	900 µm-thick PCD on Si	SAB RT	--
[114]	2019	6	Si	1 cm × 1 cm SCD	SAB RT	--
[115]	2020	10	Si	SCD	SAB RT	19 ^a^
		2				11 ^a^
[117]	2020	2 × 5/11	Mo/Au	PCD/SCD	SAB RT	--
[118]	2020	10	Ti/Si	5 mm × 5 mm SCD	SAB RT	66 ^c^
[119]	2020	2 × 450	AlN	--	SAB 160 °C	--
[122]	2018	No interlayer		SCD	VdW bonding	10 ^d^

Values obtained with ^a^ transient thermoreflectance, ^b^ on-wafer IR imaging system, ^c^ periodic heating method, and ^d^ estimated from simulations.

**Table 5 materials-15-00415-t005:** GaN epitaxy on diamond substrates.

Ref.	Year	Diamond	GaN			
			Deposition Method	Type	Thickness (µm)	Disl. Dens. (cm^−2^)
[125]	2003	Natural SCD	MOCVD on 10 nm AlN layer + HVPE	Polycryst.	2.5	--
[126]	2011	(011) SCD	MOCVD on AlN layer	Polycryst.	0.07–1.55	--
[127]	2009	(111) SCD	NH_3_-MBE on 100 nm AlN layer	Epilayer	1	--
[128,129]	2010	(111) SCD	NH_3_-MBE on 200 nm AlN + GaN strain engineered interlayers	Epilayer	0.8	8.4 × 10^9^
[130]	2011	Ib (111) SCD	MOVPE on 180 nm single crystal AlN + 400 nm AlN/GaN	Epilayer	0.6	--
[131]	2012	IIa (111) SCD	MOVPE on 180 nm single crystal AlN + 500 nm AlN/GaN	Epilayer	0.6	8.4 × 10^9^
[132,133]	2012	Ib (111) SCD	MOVPE on 180 nm single crystal AlN + 500 nm AlN/GaN	Epilayer	0.6	--
[134]	2009	NCD	MOCVD on 50 nm GaN	Polycryst.	3	--
[135]	2010	PCD	MOCVD	Polycryst.	0.8	--
[124]	2015	PCD	MOVPE on 70 nm AlN layer + deposition of SiN stripes/etching + ELO	15 µm wide epilayer	1.5	≈7 × 10^−9^ → <10^8^
[136]	2017	PCD thin films	Etching of Si substrate + MOVPE on 10–40 nm AlN/Al_0.75_Ga_0.25_N layers	Epilayer	0.2–1.1	--
[137]	2020	Post-deposited PCD	MOCVD of SiN-capped AlGaN/GaN stack on Si + selective deposition of PCD stripes + ELO	5 µm wide epilayer	≈1–5	≈10^9^ → ≈10^7^

**Table 6 materials-15-00415-t006:** Evolution of capping diamond technology.

Ref.	Year	Passivation Layer		Diamond Film			*TBR* (m^2^⋅K/GW)
		Thickness (nm)	Material	Thickness/Type (µm)	Dep. Temp. (°C)	CVD Type	
[46]	2001	--	SiN	0.7–2/PCD	<500	MPCVD	--
[141,142]	2010, 2012	50	SiO_2_	0.5/NCD	750	MPCVD	--
[143]	2013	No interlayer	--	--/NCD	750	MPCVD	--
[146]	2017	10	SiN	0.5/NCD	750	MPCVD	--
[148]	2011	--	SiO_2_/Si_3_N_4_	0.5/NCD	750–800	HFCVD	--
[149]	2014	--	Si_3_N_4_	2.8/NCD	750–800	HFCVD	--
[150]	2017	50	Si_3_N_4_	0.155–1/PCD	650	MPCVD	45 + 13/–11—91 + 13/−9 ^a^
[91]	2019	46	SiN	1.46/PCD	720–750	HFCVD	52.8 + 5.1/−3.2 ^a^
[152]	2019	30	SiN	3/PCD	820	MPCVD	--
[154]	2021	100	SiN	2.5/PCD	700	HFCVD	--

^a^ Values obtained with transient thermoreflectance.

**Table 7 materials-15-00415-t007:** Current advantages and disadvantages of each approach.

	GaN-on-Diamond	Bonded Wafers		GaN Epitaxy		Capping Diamond
		SCD	PCD	SCD	PCD	
Large area	Yes	No	Yes	No	Yes	Yes
κ_diamond_ at interface	Low	High	High	High	ELO GaN-after-PCD: high	Low
					PCD-after-ELO GaN: evaluation required	
*TBR* _GaN/diamond_	Large	Small	VdW: not reproducible	Evaluation required	ELO GaN-after-PCD: evaluation required	Large
			SAB: evaluation required		PCD-after-ELO GaN: evaluation required	
Removal of AlGaN/GaN stress-relief layers	Possible	Possible	Possible	Not possible	ELO GaN-after-PCD: not possible	Not relevant
					PCD-after-ELO GaN: possible	
AlGaN top barrier layer	Not relevant	Not relevant	Not relevant	Not relevant	Not relevant	Present
Induced thermal stress	Relevant	Not relevant	Not relevant	Not relevant	ELO GaN-after-PCD: not relevant	Relevant
					PCD-after-ELO GaN: evaluation required	
Manufacturing complexity	Fair	Fair	Fair	Simple	ELO GaN-after-PCD: fair	Simple
					PCD-after-ELO GaN: complex	

Advantage
Limitation
Severe limitation

## Data Availability

Not applicable.

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
