# Peer review of "Diamond/GaN HEMTs: Where from and Where to?"

_materials, 2022, doi:10.3390/ma15020415_

Round 1

Reviewer 1 Report

General observations:

In my opinion, the Review provides interesting data about diamonds and GaN. Overall, the article is well written and presented properly, with a pleasant read. The introduction is well written and adequately describes the work. The sections were carefully chosen and show the maturity of the work, describing the main points of these materials so important to science and industry. The figures and tables were chosen in a way that helps the reader to understand the technological evolution of the studied materials. This Review is sure to be well accepted by the community and get many citations. I say it was a pleasure to review this article as it is a pleasure to read.

Reviewer 2 Report

The authors reviewed the current state of art in the fabrication of GaN/diamond heterostructures for overcoming the thermal dissipation issues of high-power GaN devices. Various fabrication methods were covered as well as their advantages and disadvantages. The wide range and the successive descriptions of each method would be useful to receive good attention from the related field researchers. But, I would like to mention one point that may need to be addressed before accepting the paper. Authors mostly focused on describing how to fabricate GaN/diamond heterostructures, yet rarely showed the heterostructure device performance. One additional figure providing superior thermal releasing characteristics of the GaN/diamond heterostructure would greatly support the main idea of this paper.

Reviewer 3 Report

The authors present a comprehensive overview of GaN/ diamond technology that will surely be helpful in the future. It is obvious that the authors are knowledgeable and understand the concerns to thermal properties of GaN. Nevertheless, it is suggested to improve the manuscript at some key parts:

  • I would suggest to change the title slightly to include HEMTs. Not sure why “where from and where to” has to be in the title.
  • It is suggested to add a paragraph that describes the intend and structure of the review.
  • Throughout the manuscript non-quantitative descriptions such as “low temperature” etc. are used. Please use accurate numbers. Also words such as “careful” should not be used (not sure what careful means in scientific environment).
  • Since the authors are focusing on the thermal properties and transport, it is strongly suggested to add a chapter at the beginning discussing these two topics in detail. What impact thermal transport other than thermal conductivity (e.g. on page 19 interfaces are mentioned). A short review over some theoretical work would be useful but really with the focus on making the thermal transport understandable for the reader.
  • The part that is missing in all these discussions is how different approaches not only impact the thermal properties but also the actual performance of the HEMT (electrically). This should be added in each chapter accordingly.
  • The outlook is generally fine but the very final chapter 4.5 needs major additions. A conclusion of only one paragraph for such a nice review is a bit on the lower end. This is the place where the authors can summarize and present opinions and visions of their own.
  • The introduction and 3.1 need major revision.
  • A short summary of the main findings should be added after each subsection (more than “everything is summarized in table XYZ”).

In detail:

Introduction:

The introduction is very disappointing and should be completely rewritten. It is in parts incomplete, misleading, and wrong.

  • The first sentence has a few minor grammatical errors.
  • Not sure how Moore’s law is really related to GaN HEMTs. This should be revised. Si device density is in now way competing with GaN HEMTs. BTW, current Intel progress is on track to stay on track with Moore’s law (https://www.cnet.com/tech/computing/how-intel-will-keep-moores-law-cranking-for-years-to-come/). Finally, the connection of Moore’s law and high-power application is unclear.
  • “high electric field strength” = “high breakdown field”?
  • On the one hand the authors claim GaN has potential on the other hand they claim it is routinely used. They should decide what they believe. I would go with the latter.
  • When initially reading this, it was unclear that this article was focusing on HEMTs (there are other high-power devices). This should be clarified early on. (e.g., “more efficient devices”)
  • Maybe some for the confusion comes from the argument of “global warming” etc. This is really future development of GaN (and SiC) high power devices which might include some HEMTs but more likely is Schottky or pn junction based. Again, this needs to be clarified.
  • The paragraph on automobiles is displaced or unnecessary.
  • Thermal is not the only and also likely not the biggest challenges for high power applications and HEMTs. This needs a better discussion.

Overall, I would try to re-organize the introduction.

Chapter 2

This is a bit why the introduction should be better. In the introduction I was told why GaN is good for high power and then I directly learning about HEMTs. Try to make a better connection.

  • A better explanation of the HEMT is needed (it is not just a potential like a discontinuity of the bandgap as in GaAs).
  • While thermal density is high the number of 10E5 is VERY dependent on the actual device. There are many reports on significantly lower power densities.
  • The comparison with the sun is unnecessary, weird, and misleading?!
  • The impact of interfaces is missing (this is where a good discussion of thermal transport would e useful).
  • The authors mention defect densities below 10E7. What they mean is dislocation density. There are other defects (e.g., point defects >1E17 in the materials).
  • The authors should mention that <1E7 is not achieved straight forward on non-native substrates but involves HVPE growth. Even with ELO it is likely more 1E8 or so.
  • I believe graphene and graphite have similar thermal conductivity as diamond. Please revise if possible.

Chapter 3

In general, this chapter is good except for 3.1. Chapter 3.1 is distinctively different to all other chapters as it does not provide a review of the technology but rather a review of group4labs and element6 efforts. It reads more like a history of the company and maybe some other companies. University work is completely omitted and it is more a history report than a science report. The authors should make an effort to compare 3.1 to 3.2 or 3.3 and the difference is obvious.

  • Figure 2 is not necessary as it adds nothing in addition to Figure 1 and 10.
  • It is unusual to name a person by first and last name in a paper. It should be Ejeckam et al. etc.
  • Please use (i) to (vi) as mentioned in the text also in Figure 3.
  • The footnotes (1) and (2) are not adding to the discourse. Either makes these references or deleted. This goes along with the general comment on 3.1.
  • (3.2) Comments such as “technical details were not disclosed” are useless. Can the author make an educated guess? If not, the comment can be taken out.
  • (3.3) The lattice mismatch between diamond and GaN is better than that of sapphire and GaN. However, for LED’s sapphire is a standard technology. What other factors impact the growths?
  • Specially for 3.3: How do these HEMTs perform electronically?
  • “threading effects” == “threading dislocations” or better yet only dislocations (page 11 2x).
  • In Figure 9 the diamond is buried in GaN. How would that positively impact thermal transport?
  • (3.4) This technology comes from GaAs. A short (1-2 paragraphs) review on the major achievements for GaAs/diamond would be beneficial.

Chapter 4:

This chapter is generally very good. Maybe the only suggestion is not to use an abbreviation in a title (4.1.2) and to extend 4.5 significantly.

Round 2

Reviewer 3 Report

The authors have addressed all of the concerns from the first review. At this point this is an excellent work. Congratulations to the auhtors.